# Direction for Detection: A Survey of Automated Vulnerability Detection and all of its Pain Points

## Abstract

Security vulnerabilities in software can have severe consequences; however, manual vulnerability detection is costly and does not scale, especially as agentic coding frameworks increase the rate of code production. Over the last decade, a large body of research has applied machine learning machine learning to automate vulnerability detection (ML4AVD), yet self-reported performance on the most popular datasets shows no clear upward trend. The ML4AVD research community has identified several flaws in problem formulations, datasets, and metrics, but these are discussed in isolation, leaving the overarching problems that generate and reinforce these flaws unaddressed. We first systematize the field through a survey of 87 influential works based on their problem formulation, input and detection granularity, target programming languages, evaluation metrics, datasets, and detection approach. Drawing on this corpus and prior empirical work, we identify twelve pain points spanning the ML4AVD pipeline and show that they are self-reinforcing and causally inter-meshed: feedback loops between datasets, formulations, baselines, and metrics perpetuate each other and explain the field's persistent concentration on binary classification of C/C++ vulnerabilities at the function level. Thus, the field optimizes for a narrow and artificial problem that omits vulnerability type prediction, broader language support, and separation of input from detection granularity. We pair each pain point with concrete recommendations to break these loops. Finally, we use AIxCC as a case study to assess how well a recent high-profile effort aligns with these recommendations and reflect on the relevance of ML4AVD in the era of agentic AI.

## 1 Introduction

Traditionally, vulnerability detection relied on manual code reviews and security audits, both labor-intensive and time-consuming practices. However, in the past decade, researchers have developed increasingly sophisticated approaches to automate this process. Automated Vulnerability Detection (AVD) now represents a potentially transformative advancement in software security, aiming to identify and support the remediation of security flaws within codebases.

AVD approaches have evolved from rule-based methods [126, 202] to machine learning-based solutions (ML4AVD), including deep learning and large language model (LLM)-driven techniques that now dominate more recent literature [20, 21, 33, 83, 84]. Competitions such as DARPA's Cyber Grand Challenge (CGC) in 2016 [122] and the AI Cyber Challenge (AIxCC) in 2025 [120] have sought to incentivize the transformation of AVD research from prototypes to production-ready solutions. Despite this sustained effort, the performance of ML4AVD solutions does not exhibit a clear upward trend, as shown in Figure 1, which plots the self-reported $F_1$-scores from papers on three of the most common AVD datasets.

This stagnation is not incidental. The ML4AVD field has become increasingly siloed, with studies often focused on isolated tasks, limited language support, and inconsistent evaluation methodologies. A key driver is the disproportionate influence of a small number of highly-cited works that shape the choices of subsequent research: which datasets to use, which problem formulations to adopt, and which evaluation protocols to follow. Consequently, flawed assumptions embedded in foundational ML4AVD work are consistently adopted

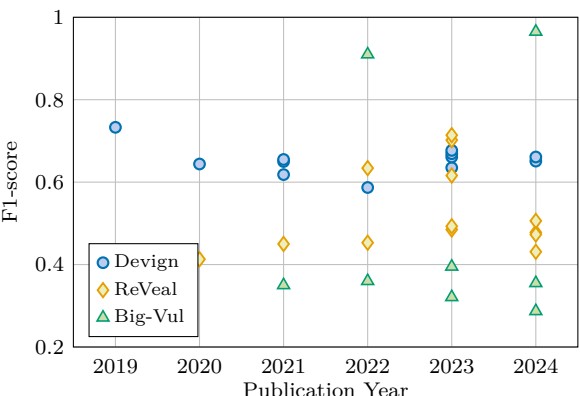

Figure 1: Self-reported $F_1$-scores by AVD solutions on the three most popular AVD datasets per year (see Appendix A for more details). Each point corresponds to one solution evaluated on a dataset. Some solutions are evaluated on multiple datasets. This illustrates some of the problems facing the ML4AVD field that we discuss in this paper. The self-reported performances for each dataset do not exhibit a clear upward trend. Even worse, due to methodological and reporting differences, comparing solutions through self-reported performance values does not provide a good picture of the state of the field.

by follow-up work and are propagate widely through the literature. This severely limits ML4AVD's broader applicability in real-world software development and misaligns research with the ultimate goal of proactive and reliable security integration.

This paper investigates the pain points that are pervasive throughout ML4AVD research and traces how influential work contributes to their persistence. To define these pain points, we collected and screened 965 articles, filtering them using a citation-frequency threshold to produce our final corpus of 87 highly-cited ML4AVD works. Using this corpus, we first systematize these articles across six dimensions: problem formulation, input and output granularity, programming language, evaluation metrics, datasets, and ML approach. We then present the key limitations across these dimensions and provide concrete recommendations for each. Our analysis is guided by the following research questions:

**RQ**. *What are the key limitations of ML4AVD research that emerge from high-cited papers in the field, how do they interact, and what do they imply for future work?*

Our analysis identifies twelve pain points spanning the ML4AVD pipeline and shows that they are not independent: they reinforce each other through feedback loops between datasets, formulations, baselines, and metrics. This integrated view explains the field's most visible symptom, namely that ML4AVD research remains highly concentrated on binary classification of C/C++ vulnerabilities at the function level, with 86% of articles using a binary formulation, 87% of single-language articles targeting C/C++, and 63% using function-level input and output granularity. Prominent function-level datasets encourage function-level solutions, which motivate the creation of additional function-level datasets; the availability of C/C++ datasets makes C/C++ the default language; and the lack of high-quality benchmarks compounds this by hindering comparability across solutions, making self-reported performance an unreliable measure of progress. The cumulative effect is a field that optimizes for a narrow and artificial task while neglecting several properties practical ML4AVD requires: predicting vulnerability type rather than mere existence, broader language coverage, and the decoupling of contextual input from fine-grained detection output.

Many of these issues have been raised individually in prior work, but their dissemination through the ML4AVD community is uneven, and their interactions remain underexplored. Existing surveys [149, 190] provide valuable taxonomies of ML4AVD research and some of its limitations, but treat pain points as independent dimensions. Our contribution is an integrated diagnosis: we show that the pain points are causally inter-meshed, that the feedback loops between them explain the state of the field better than any individual limitation, and that targeted fixes are therefore insufficient without breaking those loops. We synthesize observations from our corpus with related empirical work to provide an integrated overview of the field's key limitations alongside their motivations, implications, and recommendations.

The remainder of this paper is organized as follows. Section 2 provides background on program analysis, program representations, and the ML approaches used in AVD. Section 3 details the methodology used to collect and screen the included articles. Section 4 presents the systematization of the 87 included articles across the six core dimensions, answering RQ1. Section 5 forms the core analysis of this paper: it presents

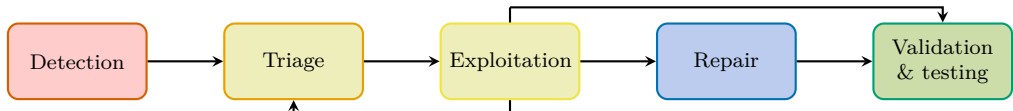

Figure 2: The stages of the vulnerability management life cycle. Arrows denote both the flow of process and contextual information.

the key limitations of ML4AVD research, referred to as the pain points, by drawing on both the quantitative findings of the previous section and the broader empirical literature, answering RQ2. Section 6 then uses AIxCC as a case study to examine how well a recent high-profile AVD effort aligns with the recommendations we provide and reflects on the relevance of ML4AVD in the era of agentic AI. Section 7 situates this work relative to existing AVD surveys. Section 9 concludes the paper.

## 2 Background

The AVD literature uses diverse input representations and program analysis methods, which have increasingly included machine learning. We briefly introduce these as a foundation for our analysis of the included articles.

### 2.1 Vulnerability Management Life Cycle

Vulnerability detection does not exist in isolation; it is the first stage in a broader vulnerability management life cycle. This life cycle, illustrated in Figure 2, mirrors industry standards such as ISO 30111 [144] and NIST SP 800-61 [192]. The five stages of vulnerability management are as follows:

**Detection:** identifying vulnerable code within the codebase. The utility of detection depends not only on *whether* a vulnerability is found but also on the richness of the information provided. Beyond a binary signal, detection can also provide the vulnerability type (e.g., CWE-ID), the affected code locations, the conditions under which the vulnerability is triggered, and the relevant code paths. This contextual information supports the subsequent stages of the vulnerability management life cycle.

**Triage:** prioritizing detected vulnerabilities based on their likely impact and urgency. Effective triage requires more than a binary detection signal, as severity depends on contextual factors such as vulnerability type, affected assets, reachability, exploitability, and potential "blast radius". In practice, exploitation can also inform triage and may be revisited as exploitation provides additional evidence about the vulnerability's practical consequences.

**Exploitation:** validating the existence of a vulnerability by constructing a reproducible exploit. Exploit construction can be informed by detection outputs, such as the triggering conditions or code paths related to the vulnerability. Although significantly more costly than detection, the exploitation phase eliminates false positives and yields a reusable test case to validate subsequent repair.

**Repair:** applying a patch to remediate the vulnerability while preserving existing functionality. Detection informs repair by localizing the affected code, while exploitation provides the test cases needed to verify that the repair is effective.

**Validation & Testing:** confirming that the patch has been applied successfully. Ensures that the vulnerability has been removed without degrading application functionality or introducing new vulnerabilities.

## 2.2    Program Analysis

Vulnerability detection is a branch of program analysis, the process of determining a program's properties. Program analysis is divided into static, dynamic, and hybrid analysis. Static analysis examines a program without executing it; thus, it can be applied to source code before compilation. Its results are generally applicable across program executions and inputs. However, static analysis of complex programs can be computationally expensive, and determining statically whether an arbitrary program is free of vulnerabilities is *undecidable* [178]. Therefore, static AVD methods require approximation to be practical, leading to diverse approaches with different trade-offs. As universal approximators [141], neural networks have been increasingly applied to this setting. As the focus of this review is vulnerability detection and we exclude exploitation, we mainly cover articles that use static analysis in the context of machine learning.

## 2.3    Program Representation

The choice of software representation determines which AVD approaches can be applied and how effective they are, as different representations preserve different information at different levels of abstraction.

**Source code:**  unprocessed human-readable instructions as plain text. Source code is the fundamental representation from which other forms are derived. ML4AVD solutions typically take a code snippet, the size and complexity of which determine the *input granularity*, to produce information about the presence and location of vulnerabilities; the level of detail at which solutions label the location of vulnerabilities determines the *output granularity*. In the literature, input and output granularities range from the project-level (the coarsest) to line-level (the finest). Input granularity presents a trade-off: coarser granularities capture more information, such as long-range dependencies, but present potential costs in training time and complexity.

**Program slices:**  subsets of code, possibly non-contiguous, that preserve the behavior of the full program with respect to a specified *slicing criterion*, such as the value of a variable at a particular program functionality [201]. In the context of AVD, slicing is used as a preliminary step to simplify further analysis by excluding unrelated code in the resulting feature space. Furthermore, *Code gadgets* [33] combine related slices and represent them as code snippets for training machine learning algorithms in AVD.

**Vectorization:**  a numeric representation of source code so it can be used as input to ML-based AVD solutions. In recent research, the most prevalent vectorization method is generating embeddings at the word, sub-word, or sentence level to capture code semantics. Embeddings can be learned from scratch through training, or directly generated from a pre-trained embedding model (e.g., `word2vec` [165], `GloVe` [173], `sent2vec` [170], and `doc2vec` [153]).

**Binary Code:**  a program representation produced by compiling the source code. It is primarily used by dynamic methods, though static analysis of binary code has been explored [75].

**Intermediate representations (IRs):**  compiler-oriented abstractions of source code, developed to support program analysis and optimization, that are now widely used beyond compilation. IRs can be instruction sets (such as bytecode) or specialized abstract data structures.

The common IR data structures in the AVD literature are:

(i) *Abstract syntax trees (AST)* capture the syntactic structure of source code; nodes represent language constructs (e.g., variables, if statements) and edges show that the parent node operates on the sub-trees rooted at the child nodes (e.g., if statements will typically connect to their condition, then-branch, and else-branch).

(ii) *Control flow graphs (CFG)* capture all paths that a program may execute; nodes represent continuous blocks of code connected by directed edges that represent jumps (e.g., if statements, function calls).

(iii) *Data flow graphs (DFG)* capture how data moves between program elements, modeling dependencies between values; nodes represent expressions that hold values and edges show the flow of data.

(iv) *Program dependency graphs (PDG)* capture both data and control dependencies in a single graph.

(v) *Code property graphs (CPG)* combine information from the AST, CFG, and PDG into a single structure.

IRs for AVD are typically generated using off-the-shelf tools, such as `ANTLRv4` [171] and `astminer` [152] for ASTs or Joern [145] for CPGs.

## 2.4   ML Approaches used for AVD

The ML4AVD literature uses a variety of ML neural network architectures:

**Convolutional Neural Networks (CNN) [154]:**   neural networks developed for grid-like data (e.g., images). CNNs apply learnable filters (called *kernels*) to sliding windows of the data and match patterns through a convolution operation. CNNs are very efficient, as the kernels' parameters are shared across dimensions, significantly reducing the number of learnable parameters.

**Recurrent Neural Networks (RNN):**   neural networks designed primarily for sequential data. The core component of an RNN is the recurrent cell. It keeps a hidden state summarizing past inputs, enabling RNNs to learn what information to retain or forget. However, basic RNNs [182] tend to easily forget inputs early in the sequence, so several improvements have been proposed, such as long short-term memory (LSTM) [140] and gated recurrent units (GRU) [118]. In addition, bi-directional RNNs [186], such as bi-directional LSTMs (BLSTM) and bi-directional GRUs (BGRU), are RNN variants in which the sequences can communicate both forward and backward. These are particularly useful when information later in a sequence can affect earlier information.

**Graph Neural Networks (GNN) [183]:**   neural networks that support learning from graph structures, where the information of each node is represented as a vector. GNNs combine messages from neighboring nodes using an aggregation function, such as sum, mean, or max. The aggregated information is then transformed to produce richer context-aware representations of nodes. The aggregation-transformation step is repeated for a number of layers to obtain representative node embeddings. GNNs can be applied to graphs irrespective of their dimensions (e.g., the number of nodes or edges).

**Large Language Models (LLM):**   transformer-based [199] neural networks that model natural language with a high number of parameters, from hundreds of millions [124] to trillions [168]. The primary innovation behind LLMs lies in the attention mechanism, which allows the model to selectively focus on different parts of the text. LLMs can be general purpose (e.g., ChatGPT [168]) or specialized in a field (e.g., CodeBERT [129]). LLMs are initially pre-trained on large text corpora in order to generate coherent and high-quality text. Pre-trained models can be adapted to a task through *fine-tuning* on a task-specific dataset.

# 3 Methodology

This survey aims to characterize the ML4AVD research landscape and explain how its recurring limitations have emerged, interacted, and persisted.

We do so in two steps: First, we systematically analyze a corpus of highly-cited ML4AVD articles to identify the methodological choices that have shaped the field, including problem formulation, input and output granularity, target programming languages, datasets, evaluation metrics, and ML approaches. Second, we synthesize these findings with the broader literature on ML4AVD pain points to examine how influential assumptions, datasets, and evaluation practices have propagated through subsequent work.

Our methodology is designed to capture work that has had measurable influence on the field, rather than provide an exhaustive census of all AVD research. We therefore collected relevant articles and passed them through multiple screening phases. To account for both influence and publication age, we applied a citation-frequency threshold of at least five citations per year. Because citation counts require time to stabilize, we excluded articles published less than one year before collection from the citation-based corpus. Newer surveys, datasets, and empirical studies are not included in the citation-filtered corpus. Instead, we incorporate them into the discussion where they help identify, corroborate, or refine the pain points and recommendations.

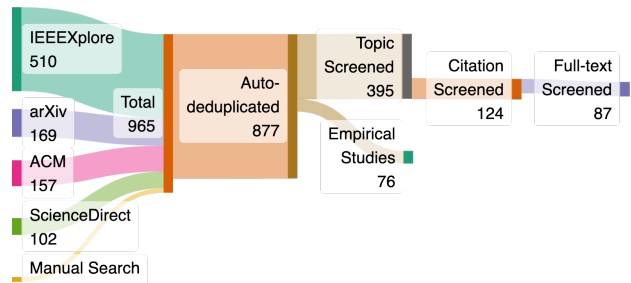

Figure 3: Article collection and screening process. 965 articles were collected from IEEEXplore (510), arXiv (169), ACM (157), ScienceDirect (102), and manual search (27). After de-duplication 877 remained; 76 were then classified as empirical studies, 395 were classified as relevant to the topic, of which 87 were included after full-text screening.

## 3.1 Article Collection

We searched four research databases that cover the vast majority of research venues in both the security and machine learning fields: *IEEEXplore*, *ACM Digital Library*, *ScienceDirect*, and *arXiv*. This methodology ensures a broader and more representative sample than including only articles from a list of selected top venues [176], as done by other systematic reviews [185]. We applied the following search criteria: *(i) The article title must include either "vulnerability detection" or "vulnerability discovery", and (ii) The body of the article must include "software".* To include relevant articles that may have been missed by the above search or were not available in the considered databases, we conducted a manual search based on examining existing AVD surveys. The cutoff is September 2024 to allow for the citation count, which we use in subsequent filtering, to stabilize. We include articles published later but with pre-prints made available before the cutoff date, as it is common for pre-prints to gain citations even prior to publication in a peer-reviewed venue. The collection process resulted in 965 articles, with the sources listed in Figure 3, which also summarizes the screening process.

## 3.2 Article Screening

To ensure the relevance and impact of the included articles, we performed three rounds of screening: topic-based, citation-based, and full-text screening.

**Topic-Based Screening.** We used *Rayyan* [169] to conduct automatic article de-duplication, as well as title and abstract-based screening. Articles were included if they satisfied the following criteria: (i) Proposed a novel AVD solution (excluding surveys, empirical studies, or datasets). (ii) Focused on detection, not exploitation or repair. (iii) Addressed security vulnerabilities, excluding syntactic or semantic bugs. (iv) Targeted vulnerabilities in software applications, not communication networks or smart contracts. (v) Excluded

detection of malicious software. (vi) Were full papers; not theses, posters, or tutorials. Using these criteria, we reduced the selection to 395 articles and 76 empirical studies.

**Citation-Based Screening.** To include only articles with a significant impact in the field, we further filter the above 395 articles based on citation count adjusted for their age to fairly compare research of varying ages [132]. We calculate citation frequency $R_c$ for each article as $R_c = \frac{N_c}{A}$, where $N_c$ is the number of citations achieved by an article as recorded in September 2025, and $A$ is the article's age in years from first publication or pre-print. The age is computed as $A = Y_n - Y_p$, where $Y_n = 2025.67$ is the time of citation re-collection (corresponding to September 2025), and $Y_p$ is the year of publication for the article. As cited computer science articles had a mean of five citations after one year from publication [130], we set a threshold of $R_c \geq 5$, resulting in 124 articles.

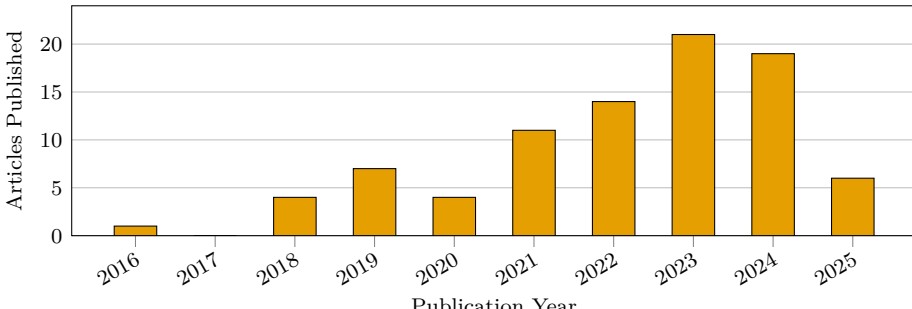

Figure 4: Distribution of the publication year for included papers after the full-text screening phase. Papers published in peer-reviewed venues after the September 2024 cutoff are listed with the updated publication year, not the date of the pre-print.

**Full-Text Screening.** The third and final screening phase is based on the full text of the remaining 124 articles. In this phase, we manually excluded duplicate articles that were not automatically detected by *Rayyan*, articles deemed irrelevant based on the full text (e.g., exploitation solutions without a detection component or articles not using ML techniques). Figure 4 presents a histogram illustrating the distribution of the 87 included articles published between 2016 and 2024. Although older articles have had more time to accrue citations, the publication-year distribution does not suggest that the citation-frequency threshold systematically favored older work.

## 4    Systematization of AVD Literature

To inform our discussion of identified limitations, we first characterize the 87 included ML4AVD articles according to six key dimensions: problem formulation, input and output granularity, programming languages, evaluation metrics, datasets, and ML approaches. For each dimension, we report the distribution of choices made across the corpus. Papers may appear in multiple categories if they meet the criteria for each category.

### 4.1    Problem Formulation

All studied articles viewed AVD as a classification problem, with two prevalent formulations. (i) *Binary*: the ML4AVD solution returns whether the input software is vulnerable or not, without specifying the vulnerability type. (ii) *CWE-ID*: the ML4AVD solution returns the type of vulnerability (often represented by the Common Weakness Enumeration identifier, or CWE-ID) to which the input software is vulnerable, if any. The ML4AVD solution could either be one multi-class classifier with multiple CWE-IDs as classes or an ensemble of binary CWE-specific classifiers.

The distribution of the problem formulations can be seen in Table 1 and Figure 5. The table shows that the *binary* formulation dominates the literature with 86% of the included articles. However, this dominance is not uniform over time, as CWE-based formulations become more prevalent starting in 2022, suggesting an emerging shift toward more informative problem formulations in more recent work.

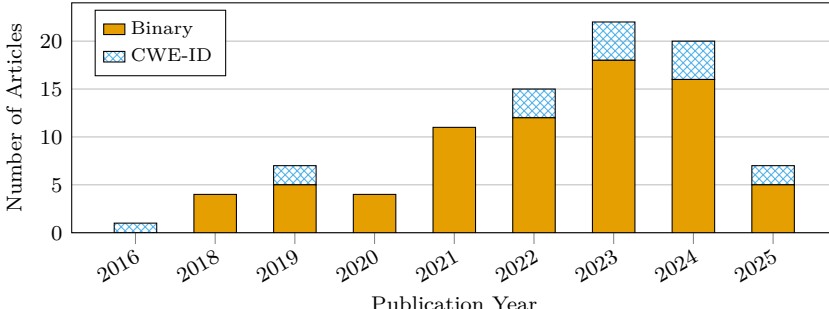

Figure 5: Distribution of AVD problem formulations over time. Binary classification dominates consistently throughout the surveyed period. CWE-ID formulations remain rare but shows growth from 2022 onwards. Articles using both formulations are counted in each.

Table 1: Classification of included articles according to the AVD problem formulation.

| Formulation | No. | Articles |
|---|---|---|
| Binary Classification | 75 | [1, 3, 4, 6–33, 36, 38–42, 44–49, 52–54, 56–68, 70–74, 76–86] |
| CWE-ID Classification | 16 | [2, 5, 21, 24, 26, 34, 35, 37, 43, 50, 51, 55, 59, 69, 75, 87] |

## 4.2 Input and Output Granularity

ML4AVD solutions take a unit of code, the size and complexity of which determine the **input granularity**. To produce information about the presence and location of vulnerabilities; the level of detail at which solutions label the location of vulnerabilities determines the **output granularity**. In the analyzed literature, input and output granularities range from the project-level (the coarsest) to line-level (the finest).

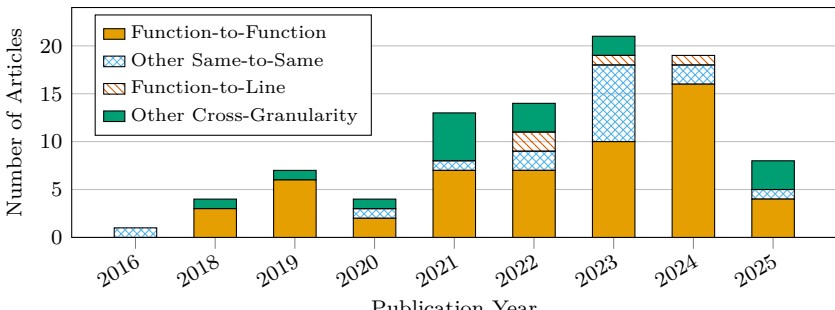

Figure 6: Distribution of input-to-output granularity configurations over time. Function-to-function and other same-to-same dominates consistently throughout the surveyed period.

Figure 6 and Table 2 show the distribution of the input and output granularities in the included articles. A vast majority (63%) of the solutions provided function-level labeling of function-level inputs. Beyond the function-level, this same-to-same relationship between input and output extends to other granularities, accounting for 82% of all configurations. Works with line-level output are rare; only four articles produce line-level output, all published in 2022 or later [7, 16, 20, 23]. Two notable articles [8, 32] use the coarsest input granularity, project-level, paired with the finest output granularity, line-level.

At the coarser end, commit-level solutions are represented by only three articles [30, 34, 45]. Slice-level solutions, which use non-contiguous code segments related to specific functionality, represent a middle ground between function and line level; six articles use slice-to-slice configurations [11, 19, 24, 25, 69, 82]. Project-level inputs are predominantly associated with slice-level outputs [2, 12, 31, 33, 35, 87], demonstrating the use of program slicing to refine the granularity of the code.

Table 2: Classification of included articles according to input and output granularity.

| Input | Output | No. | Articles |
|---|---|---|---|
| Project/executable | Project/executable | 2 | [26, 51] |
| | Function | 1 | [75] |
| | Slice | 6 | [2, 12, 31, 33, 35, 87] |
| | Snippet | 1 | [72] |
| | Line | 2 | [8, 32] |
| Commit | Commit | 3 | [30, 34, 45] |
| File | Line | 1 | [44] |
| Function | Function | 55 | [1, 3–6, 9, 13–15, 17, 18, 21, 22, 24, 27–29, 36–40, 42, 43, 46–50, 52, 53, 55–60, 62, 63, 65–68, 70, 71, 74, 78–86] |
| | Line | 4 | [7, 16, 20, 23] |
| | Slice | 3 | [24, 77, 82] |
| | Snippet | 2 | [54, 81] |
| Slice | Slice | 6 | [11, 19, 24, 25, 69, 82] |
| Snippet | Snippet | 5 | [10, 41, 61, 64, 73] |

## 4.3 Programming Languages

The included articles show a striking lack of diversity in the programming languages they target, with the vast majority focusing on a single language. Figure 7 and Table 3 show the distribution of the programming languages targeted by the included literature. The literature is dominated by works looking at a single language (92%), of which C/C++ is the overwhelmingly most common target (87% of single-language articles). Additionally, all articles that consider multiple languages include C/C++ as one of them. The dominance of C/C++ is not a recent phenomenon as it is present across the entire surveyed period. This demonstrates that the growth of the ML4AVD field has not been accompanied by any meaningful diversification of language coverage; rather, the expanding literature has largely maintained the same focus.

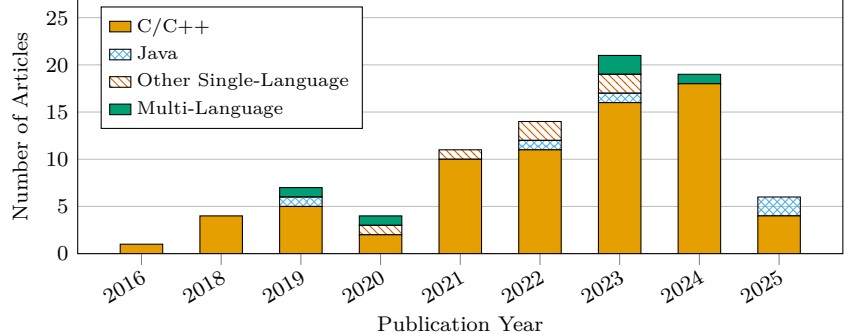

Figure 7: Distribution of targeted programming languages over time. C/C++ dominates consistently throughout the surveyed period. Multi-language and non-C/C++ single-language articles remain rare throughout.

Non-C/C++ single-language articles appear only sporadically, with Java representing the most consistent alternative [2, 35, 43, 44, 50] with other languages primarily being represented by only a single article across the entire corpus [19, 41, 64]. Multi-language articles, though present in small numbers from 2019 onward, show no sustained upward trend, remaining at one or two articles per year at most.

Table 3: Classification of included articles according to the targeted programming languages.

| No. of languages | No. | Language(s) | No. | Articles |
|---|---|---|---|---|
| Single language | 82 | C/C++ | 71 | [1, 3–9, 11–18, 20–26, 28, 29, 31–34, 36–40, 42, 45–49, 52–59, 61–63, 65–71, 73, 74, 76, 77, 79–87] |
| | | Java | 5 | [2, 35, 43, 44, 50] |
| | | Assembly/binary | 3 | [51, 72, 75] |
| | | JavaScript | 1 | [41] |
| | | PHP | 1 | [19] |
| | | Python | 1 | [64] |
| Multi-language | 5 | C/C++, Java | 2 | [30, 78] |
| | | C/C++, Python | 1 | [27] |
| | | C/C++, Java, Swift, PHP | 1 | [60] |
| | | C/C++, JavaScript, Ruby, Go, Java, C#, Python | 1 | [10] |

## 4.4   Datasets

The included AVD articles exhibit wide variability in their evaluation datasets. The majority (59.7%) either created new datasets from scratch or extracted subsets of openly-available vulnerability databases; 49.9% used existing benchmark datasets (one article may use both dataset types). Out of the 13 identified datasets, the nine most frequently used evaluation benchmark datasets are shown in Table 4 and Figure 8. The three most used datasets (Devign [84], ReVeal [9], and Big-Vul [92]) together account for the evaluation in a substantial proportion of the corpus, with Devign [84]/CodeXGLUE [97] alone used in 27 articles.

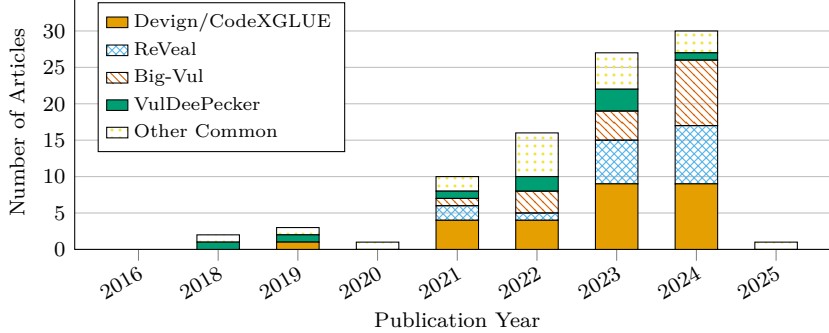

Figure 8: Usage of benchmark datasets over time. Devign/CodeXGLUE, ReVeal, and Big-Vul account for the majority of benchmark usages. "Other Common" combines SySeVR, Draper, D2A, and $\mu$VulDeePecker.

All of the most common datasets target C/C++ code, and all are at either the function or slice level of granularity, directly reflecting the trends observed in subsection 4.2 and subsection 4.3. Big-Vul is the only dataset in the top nine to provide labels at more than one granularity, including both function-level and line-level annotations. Furthermore, two of the most widely used datasets (Devign [84]/CodeXGLUE [97] and ReVeal [9]) do not provide CWE-IDs, which may directly contribute to the observed predominance of binary classification in the literature observed in subsection 4.1.

Function-level datasets are typically built by extracting code changes from security-related commits in open-source projects (e.g., *Devign* [84]: FFMPeg+Qemu, *Reveal* [9]: Linux Debian kernel and Chromium, *Big-Vul*: 348 projects, *D2A*: OpenSSL, FFMpeg, httpd, NGINX, libtiff, and libav). Whereas most of the slice-level datasets [31, 33, 87] are extracted from two sources maintained by the US National Institute of Standards and Technology (NIST): the National Vulnerability Database (NVD) [94], containing real-world programs, and the Software Assurance Reference Dataset (SARD) [95], containing both real and synthetic programs.

Table 4: Classification of included articles according to the dataset used.

| Dataset | No. Articles |
|---|---|
| Devign [84]/CodeXGLUE [97] | 27 [1, 7, 9, 13, 21, 28, 29, 40, 42, 45–48, 58, 61, 62, 65–68, 76, 79–82, 84, 85] |
| ReVeal [9] | 17 [7, 9, 21, 28, 29, 42, 57, 59, 62, 65–68, 76, 79–81] |
| Big-Vul [92] | 17 [7, 13, 20, 23, 29, 30, 42, 45, 47, 48, 53, 62, 65, 67, 68, 73, 81] |
| VulDeePecker [33] | 9 [11, 18, 21, 25, 26, 33, 59, 72, 87] |
| SySeVR [31] | 6 [18, 24, 31, 77, 78, 82] |
| Draper [49] | 5 [1, 18, 21, 49, 85] |
| D2A [104] | 5 [13, 21, 59, 69, 79] |
| $\mu$VulDeePecker [87] | 4 [18, 21, 59, 87] |

## 4.5 ML Approaches

Among our included articles, the most common ML approaches used to solve AVD are CNNs, RNNs, GNNs and LLMs, all of which are sub-branches of deep learning. Table 5 shows the full distribution, and Figure 9 shows the evolution of approach usage over the surveyed period.

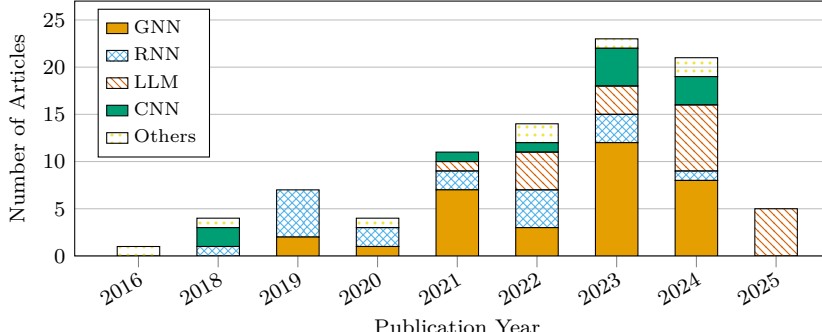

Figure 9: Distribution of the most-common six AVD detection approaches over time. RNNs dominate during 2017-2020, then GNNs during 2021-2023, and finally a large shift to LLM and transformer-based solutions in 2024-2025.

RNNs were the dominant approach from 2017 to 2020, reflecting their established capability for modeling sequential data. Among RNN variants, bi-directional configurations (BLSTM and BGRU) are the most common, as propagating information in both directions through a code sequence enables the capture of dependencies that are missed by unidirectional models.

GNNs superseded RNNs as the dominant approach from 2021 to 2023 and represent 37.9% of all included articles. GNNs are well-suited to AVD through their ability to capture the structural properties of source code through graph-based intermediate representations (ASTs, CFGs, PDGs, CPGs), enabling detection approaches to reason about program structure. Within GNNs, graph convolutional networks (GCN) and gated graph neural networks (GGNN) are the most common variants, followed by graph attention networks (GAT) and relational GCNs (RGCN).

Finally, LLMs account for 23.0% of articles overall but represent 46.2% of the approaches in 2024 and 2025 articles, demonstrating a sharp and recent shift in the field. LLM-based solutions are applied either through prompting (12 articles) or fine-tuning (8 articles), with one article combining pre-training and fine-tuning. The attention mechanism underpinning LLMs supports the capture of long-range dependencies in code, and pre-training on large code corpora provides a strong foundation for downstream vulnerability detection tasks.

Table 5: Distribution of the approaches used for AVD in the included papers.

| Approach | No. | Sub-approach | No. | Articles |
|---|---|---|---|---|
| GNN | 33 | GCN | 10 | [14, 16, 30, 41, 45, 58, 63, 67, 82, 85] |
| | | GGNN | 7 | [9, 28, 37, 53, 60, 69, 84] |
| | | GAT | 5 | [23, 47, 70, 79, 80] |
| | | RGCN | 4 | [16, 45, 67, 82] |
| | | Others | 11 | [6–8, 12, 18, 26, 29, 46, 57, 65, 76] |
| RNN | 18 | BLSTM | 9 | [3, 27, 33, 36, 38, 39, 41, 54, 87] |
| | | LSTM | 7 | [1, 19, 50, 51, 63, 64, 75] |
| | | BGRU | 2 | [31, 32] |
| LLM | 20 | Prompting | 12 | [2, 5, 10, 17, 35, 40, 42, 44, 48, 56, 78, 83] |
| | | Fine-tuning | 8 | [10, 20, 24, 25, 52, 62, 73, 86] |
| | | Pre-training & Fine-tuning | 1 | [21] |
| CNN | 11 | - | 11 | [3, 4, 11, 22, 47, 49, 55, 61, 71, 72, 81] |
| Others | 8 | - | 8 | [13, 15, 34, 43, 59, 66, 68, 77] |

## 4.6 Evaluation Metrics

The evaluation metrics used across the included articles reflect the predominance of binary classification as the standard AVD problem formulation. Figure 10 shows the most commonly used metrics in the analyzed literature. The overwhelming majority of the works use the typical metrics for classification: *Accuracy*, *Precision*, *Recall*, False Positive Rate (*FPR*), and False Negative Rate (*FNR*). To capture the interaction between *Precision* and *Recall*, the $F_1$-score is also frequently reported and is the single most common metric, being used in 68 of the 87 articles; much less frequently, the Precision-Recall Area Under Curve (*PR-AUC*) is used in two included articles. Unbiased metrics, such as the Matthews correlation coefficient (MCC), which is an unbiased counterpart of $F_1$, and Informedness and Markedness, which are unbiased versions of *Recall* and *Precision*, respectively [175], are reported rarely. The Receiver Operating Characteristic Area Under Curve (*ROC-AUC*) is reported in seven works to show the trade-off between *Recall* and *FPR*.

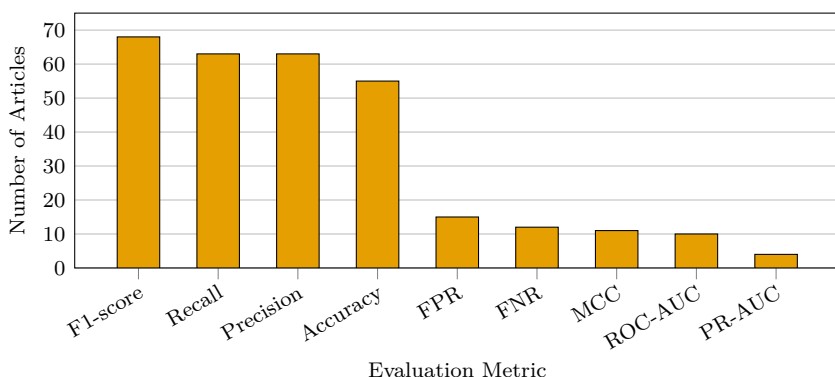

Figure 10: Distribution of the most common evaluation metrics used in AVD. F1-score, Recall, and Precision are the most widely reported, while area-under-curve metrics remain less common.

## 5 Pain Points and Recommendations

In this section, we identify the pain points and limitations of ML4AVD based on the included literature and provide recommendations to address them. Building on the quantitative findings of section 4, we qualitatively examine the motivations behind these limitations as articulated in the included papers and trace how they propagate through the field. While many of these issues have been raised individually in prior work, no existing source provides an integrated view.

Articles positioned as guiding practice (e.g., [194]) do not cover these recurring pitfalls, and works that do address them tend to be narrow in scope or focused on a single dimension of the problem (see section 7). As a result, the dissemination of these insights through the ML4AVD community remains uneven, and known limitations continue to resurface in new work. Following the approach of Arp et al. [108], we synthesize knowledge from disparate sources to provide a comprehensive picture of the pain points of ML4AVD, the interactions between them, and concrete directions for moving beyond them.

The pain points are grouped roughly following the main topics introduced in section 4: **research question** (cf. subsection 4.1 and subsection 4.2), **programming languages** (cf. subsection 4.3), **datasets** (cf. subsection 4.4), **training** (cf. subsection 4.5), and **evaluation** (cf. subsection 4.6).

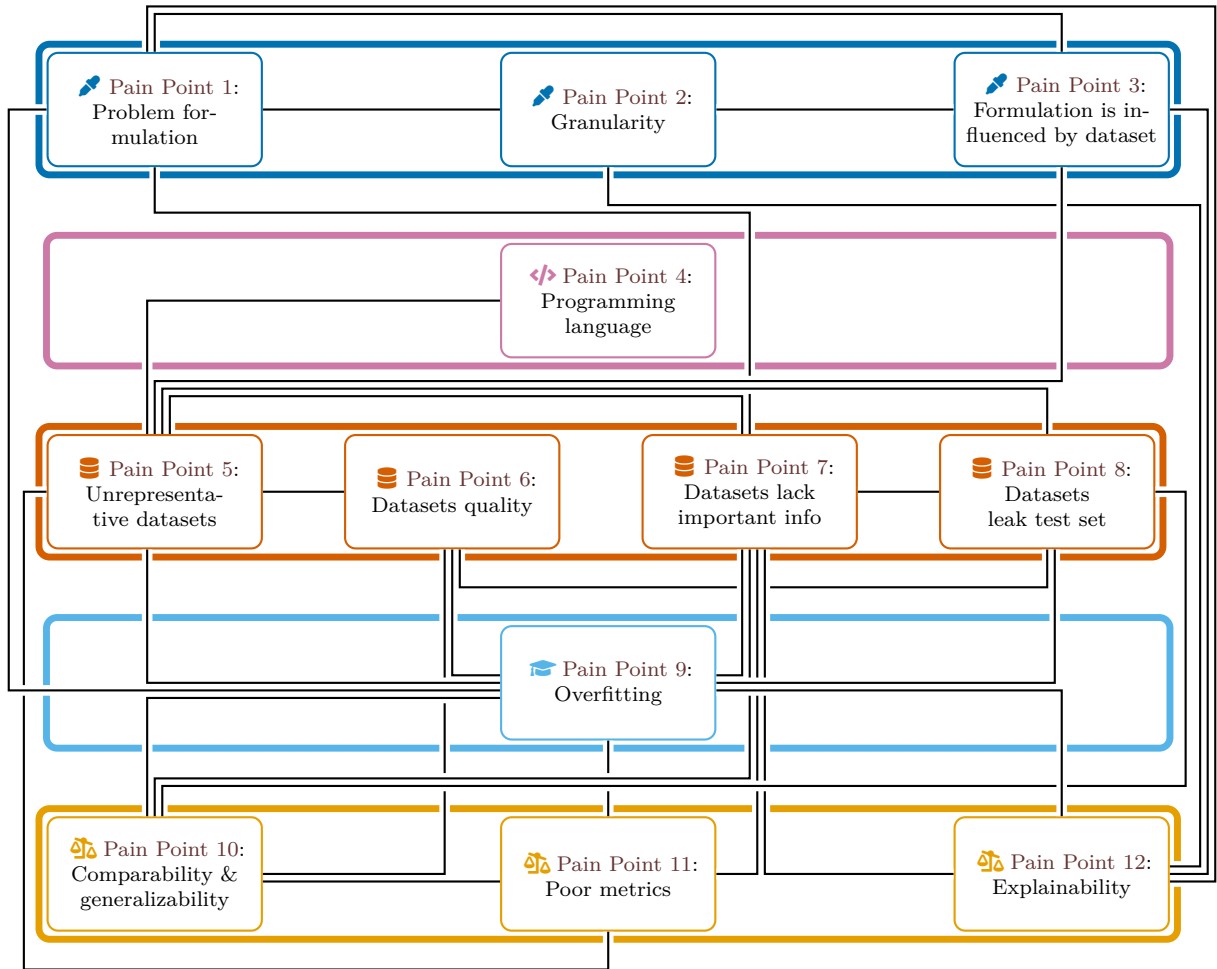

Figure 11: Relationships between pain points grouped into **research question**, **programming languages**, **datasets**, **training**, and **evaluation**.

---

📌 **Pain Point 1: Problem formulation focuses on binary classification**

AVD is conceptualized as a standalone classification task, with a large focus on binary labels; this limits the value of the research in the wider context of the vulnerability life cycle.

**Description and Motivations.**   As observed in subsection 4.1, all included articles explicitly or implicitly frame AVD as a classification task. On the rare occasion this choice is motivated, it is justified with reference to classification being the prevailing formulation in the field (e.g., "we followed the problem formulation used by Russell et al. [49]" and "Most of the approaches formulate the problem as a classification problem" [9]). In this context, binary classification of code into vulnerable or benign classes is considered the de facto default. Articles studying binary AVD classification rarely explicitly formulate this (e.g., [28, 74]); when they do, they provide no motivation for this formulation, while classification by CWE-ID is left as future work (e.g., [73]). In contrast, articles that adopt a multi-class formulation motivate it as an improvement over binary classification, as each vulnerability type will have different features [35, 69]. A few articles go beyond classification and conduct small user studies on the usefulness of their ML4AVD solution for subsequent stages in the vulnerability life cycle (e.g., [7, 17, 32]).

**Implications.**   Vulnerability detection appears as a standalone task in much of the analyzed ML4AVD literature. In practice, it is only the first step of the vulnerability life cycle and feeds into a broader process. Although binary formulations are simpler to solve, simply classifying a unit of code as vulnerable does not provide the information required by subsequent steps. Providing the vulnerability type (CWE-ID) supports triage, replication, and repair, whether performed automatically or by a human expert. For this reason, most vulnerability detection tools currently used in industry provide the vulnerability type [167]. For example, vulnerability scanners such as Nessus [196] and OpenVAS [135] use Common Vulnerabilities and Exposures (CVE) identifiers, while static analysis tools such as SonarQube [150] and CodeQL [134] use labels akin to CWE-ID types (CWE, CVE, or OWASP). The continued focus on binary classification thereby illustrates the misalignment between AVD research and real-world AVD. Nevertheless, 10 of the 16 articles using CWE-based formulations were published in 2023 or later, suggesting an emerging interest in this more practical formulation.

**Recommendations.**   The problem formulation of ML4AVD should take into account the vulnerability life cycle and the role detection plays in it. Thus, it should seek to provide rich context that can inform downstream steps of the vulnerability life cycle. At a minimum, multi-label classification based on the CWE-ID (or similar) provides information on severity required for triage, which also allows algorithms to learn features specific to each type of vulnerability. Moving beyond the narrow scope of classifying a unit of code, novel formulations should be explored, as recommended by Risse and Böhme [180]. Additional information for each vulnerability, such as conditions and code paths that can trigger it, would be valuable for triaging the severity of the vulnerability and replicating it. Knowing the context under which a vulnerability is triggered also helps produce tests to validate repairs and prevent the vulnerability from resurfacing (i.e., regressions). This aligns the research with real-world practices and enables more practical evaluation (see also Pain Point 10).

To achieve this, using representations of code that better capture context and long-range dependencies, such as program slices and graph IRs, can help provide this additional context (see also Pain Point 2). Furthermore, program analysis techniques such as symbolic execution [151], concolic testing [187], and guided fuzzing [164] are complementary to ML4AVD. Providing additional information is also covered in Pain Point 12, which discusses interpretability and explainability.

---

📌 **Pain Point 2: Granularity does not differentiate between input and output**

ML4AVD overwhelmingly uses the same granularity for both input and output, with function-to-function approaches being the majority of the field as a compromise between context and precision.

**Description and Motivations.**   The majority of articles we analyzed operate on the same granularity for both input and output, as noted in subsection 4.2. This appears to be the 'default' in the literature and

indicates that input and output granularities are conflated (e.g. [36, 70]). Function-level granularities are argued to offer a useful level of detail while still retaining enough context [1, 49, 84], though this is obviously not always the case, as vulnerabilities that span multiple functions are sometimes discarded, as in [39], and line-level detection is left as future work in [53, 64]. Therefore, it is not surprising that coarser granularities, e.g., commit and slice, are motivated by the need for additional context for classification [45, 54]. In contrast, line-level output granularities are motivated as being more precise and usable [20, 23, 32].

**Implications.** A same-to-same relationship introduces an artificial trade-off: an increase in granularity that captures more contextual information would produce less specific output and vice-versa. Additionally, the most common granularity, function-level, offers an imperfect compromise. It does not capture enough information and prevents using techniques that encode relationships (e.g., graph representations, slicing) that occur across multiple functions. Real-world vulnerabilities frequently occur at inter-procedural boundaries where function-level solutions struggle [156, 58] and a manual expert review of function-level datasets found that vulnerabilities are overwhelmingly context-dependent [180], most commonly on argument values or the behavior of other functions that are not included in the sample. Function-level granularity also lacks the precision to target subsequent repair, as functions may be arbitrarily complex and thus require significant additional effort to patch the vulnerable code.

**Recommendations.** Future work should separate input and output granularities.

*For inputs*, solutions should include all required context, therefore, targeting project-level input granularity, which is known to produce better results compared to function-level [103]. To capture long distance, inter-procedural dependencies, graph-based IR can be used alongside other representations.

*For outputs*, solutions should aim for fine granularity to best inform subsequent steps in the vulnerability life cycle, pointing to one or more specific lines of code and the code paths and conditions required to trigger the vulnerability (as noted in Pain Point 1). This is akin to slicing, however, as noted by Risse and Böhme [180], purely relying on program slicing delegates the problem to slicing with the correct criterion, rather than solving it. Note that solving at a low granularity implicitly solves all higher granularities, as a function containing a vulnerable line is itself vulnerable, and so on, with the additional benefit of better contextualizing any findings. Finer granularity implicitly provides better explainability to the output of ML4AVD models. We discuss explainability further in Pain Point 12.

Future work should look to use multi-granularity datasets, such as ReposVul [102] or VulEval [103]. If using function-level datasets that include real-world data, additional information, such as repository and commit id, can be used to reconstruct project-level inputs. For example, the Real-Vul dataset [89] is constructed from the Big-Vul dataset [92] but at the repository-level. Additionally, having access to commit history and non-code data (see also Pain Point 7) to capture the information required to decide on vulnerabilities in complex projects can improve results, especially when using LLM-based methods [17, 73]. Recommendations on how datasets can support this future work are discussed in Pain Point 3 and Pain Point 7.

> 📌 **Pain Point 3: Problem formulation is influenced by available dataset**
>
> The availability of function-level datasets and the focus of influential papers on binary classification has implicitly pushed the field into the most common formulation: function-to-function binary classification.

**Description and Motivations.** The patterns discussed in Pain Point 1 and Pain Point 2 arise in part due to the availability of datasets designed for function-level (binary) classification. The early focus on functions-to-function classification has produced many prominent function-level datasets (e.g., [9, 49, 84]). The choice of dataset in the analyzed literature is sometimes motivated by existing high usage in the field (e.g., [40, 66]). Creating high-quality datasets requires large time-investments (e.g., "600 hours" [84] or "150 hours" [180]), thus using available datasets is a significant time-saving choice. It also enables comparison with preceding work, though the problems with baselines are explored in Pain Point 10. Lack of available datasets is cited as a reason for coarse-grained detection [7] and limited line-level baseline comparisons [20].

**Implications.**   Popular datasets shape the direction of the field by making certain problem formulations more tractable to study, while prior work reinforces those formulations by motivating their use and providing established points of comparison. In turn, this further propagates the datasets, leading to a feedback loop that reinforces existing paradigms.

**Recommendations.**   Datasets should attempt to be formulation-agnostic. In support of the recommendations from Pain Point 1 and Pain Point 2, datasets need to contain full context, preferably (a way to reconstruct) repositories with version control history, and provide rich multi-class labeled data from project-level down to line-level. For work that consumes existing datasets, we recommend authors critically evaluate datasets and the influence their limitations might have on the work [108, 127]. Additionally, information about the CWE, explanations of the vulnerability, and conditions under which it is triggered may be useful in extending the task beyond pure classification. This additional data is further explored in Pain Point 7.

Many of the subsequent pain points are also dataset-related. Much of the work that proposes new datasets is done in response to perceived flaws in existing datasets and some explicitly refines previous datasets. To better capture these pain points, we extract datasets used in the analyzed research (Table 4) and add datasets proposed by related work or discovered through snowballing to produce Table 6. The table presents the specific set of dataset attributes that relate to the pain points discussed in this section and illustrates the distribution of dataset-related pain points. For a broader discussion of datasets, we refer readers to the survey by Zhu et al. [208].

---

**</> Pain Point 4: Programming languages are not proportional to use or susceptibility**

C/C++ is overrepresented compared to real-world usage or the prevalence of vulnerabilities, in part, influenced by pre-processing tools.

---

**Description and Motivations.**   Subsection 4.3 shows that the included literature mainly focuses on solutions that analyze a single language, most commonly C/C++. As with previous pain points, this appears to be the 'default' in ML4AVD research, rarely being explicitly motivated. The availability of C/C++ datasets is one explanation [77, 81], as is the focus on memory-related vulnerabilities [8]. Due to the explicit memory management model, C and C++ are notorious for memory-based vulnerabilities, which make up 40% of the MITRE 2023 top-25 dangerous software weaknesses [166] Additionally, for GNN-based solutions, the languages that tools used to generate the graph-based IRs (e.g, Joern [145]) are cited as a factor in the choice of language [6, 16, 55, 59, 70, 85]. Solutions are claimed to be applicable across languages [14, 16, 22, 34, 49, 53, 56, 59, 63] but not evaluated or left as "future work" [3, 10, 11, 62, 77, 82, 83], though some works acknowledge that work would be required to adapt to another language [31, 76, 79]. Where other languages are analyzed, this is explicitly in contrast to the focus on C/C++ [64].

**Implications.**   Programming languages vary in susceptibility to vulnerabilities and in prevalence in software development. Therefore, we conduct a comprehensive, multi-faceted analysis of how the current AVD research landscape maps to the real-world with regard to programming languages. We consider the popularity of languages in real-world software development, and the relative susceptibility of a language to vulnerabilities.

Figure 12 compares the languages across multiple metrics. The first metric is the distribution of languages among ML4AVD research, as presented in subsection 4.3. The second metric is the proportion of MITRE's 2023 Top-25 dangerous software weaknesses [166] to which a language is vulnerable. The top languages (C/C++, Java, and PHP) are all equally vulnerable.

However, some vulnerabilities are much more prevalent in real-world software. MITRE's Top-25 weaknesses are determined using a *danger score* that combines the detection frequency of a vulnerability and its severity. Therefore, the Top-25 inherently includes prevalence, but the score discrepancy is wide, with scores of more than 60 and less than 4. This motivates our third metric, which captures the overall severity of a language's vulnerabilities. For each language, we compute a *combined danger score* by adding the danger scores of its vulnerabilities, then normalizing it by the sum of all the Top-25 vulnerabilities. From this, PHP is the most

Table 6: Comparison of vulnerability datasets.

| Dataset | Cit. | Year | Lang. | Size[1] ('000) | % Vuln. | Real world | Gran.[2] | CWE ID | T/t[3] split | Code pairs[4] | Metadata |
|---|---|---|---|---|---|---|---|---|---|---|---|
| Devign | 84 | 2019 | C/C++ | 27 | 45.0 | ✓ | f | ✗ | ✗ | ✗ | Repo, commit |
| CodeXGLUE[5] | 97 | 2019 | C/C++ | 27 | 45.0 | ✓ | f | ✗ | ○[6] | ✗ | Repo, commit |
| VPP[7] | 100 | 2024 | C/C++ | 26 | 50.0 | ✓ | f | ✗ | ○[6] | ✓ | Repo, commit |
| ReVeal | 9 | 2021 | C/C++ | 18 | 9.2 | ✓ | f | ✗ | ✗ | ✓ | Repo |
| Big-Vul | 92 | 2020 | C/C++ | 189 | 5.7 | ✓ | f, L | ✓ | ○[8] | ✓ | Repo, commit, bug report, CVE |
| VulDeePecker | 33 | 2018 | C/C++ | 62 | 28.8 | ○[9] | S | ✓ | ✗ | ✗ | |
| SySeVR | 31 | 2018 | C/C++ | 421 | 13.4 | ○[9] | S | ✓ | ✗ | ✗ | |
| Draper | 49 | 2021 | C/C++ | 1274 | 9.5 | ○[9] | f | ✓ | ○[6] | ✗ | |
| μVulDeePecker | 87 | 2019 | C/C++ | 182 | 23.7 | ○[9] | S | ✓ | ✗ | ✗ | |
| D2A | 104 | 2021 | C/C++ | 1296 | 1.4 | ✓ | S | ✓ | ○[6] | ✓ | Repo, commit |
| CVEfixes | 88 | 2021 | 27 lang. | 139 | 50.0 | ✓ | F, f | ✓ | ✗ | ✓ | Repo, commit, CVE |
| CrossVul | 99 | 2021 | C/C++, JS, PHP, Py, Ruby | 27 | 50.0 | ✓ | F | ✓ | ✗ | ✓ | Repo, commit, CVE |
| SVEN[10] | 93 | 2023 | C/C++, Py | 2 | 50.0 | ✓ | f | ✓ | ○[6] | ✓ | Repo, commit |
| DiverseVul | 90 | 2023 | C/C++ | 349 | 5.4 | ✓ | f | ✓ | ✗ | ✗ | Repo, commit |
| Real-Vul | 89 | 2024 | C/C++ | 1688 | 0.3 | ✓ | R, f, l | ✓ | ✓ | ✗ | Repo, commit |
| MegaVul | 98 | 2024 | C/C++, Java | 340 | 5.4 | ✓ | f | ✓ | ✗ | ✓ | Repo, commit, CVE |
| ReposVul | 102 | 2024 | C/C++, Java, Py | 262 | n/s | ✓ | R, F, f, l | ✓ | ✗ | ✓ | Repo, commit, CVE |
| PrimeVul[11] | 91 | 2024 | C/C++ | 236 | 3.1 | ✓ | f | ✓ | ✓ | ✓ | Repo, commit, CVE |
| VulEval | 103 | 2024 | C/C++ | 232 | 3.0 | ✓ | R, F, f | ✓ | ✓ | ✗ | Repo, commit, dependencies, CVE |
| CleanVul | 96 | 2025 | C/C++, Py, JS, Java, C# | 16 | 50.0 | ✓ | f | ✓ | ✗ | ✓ | Repo, commit, CVE |
| PairVul | 17 | 2026 | C/C++ | 6 | 50.0 | ✓ | f | ✓ | ○[6] | ✓ | CVE |

[1] Size in thousands; where multiple granularities are present, this shows number of functions for easier comparison with the dominant granularity.

[2] Granularity values in this column represent Line (L), Slice (S), Function (f), File (F) or Repository (R).

[3] Fixed train/test/validate split.

[4] Explicitly providing both vulnerable and patched code for each datapoint; datasets that provide enough information (e.g., repository and commit) allow this to be reconstructed are not included.

[5] Contains the Devign dataset but with randomly-allocated fixed train/test splits.

[6] Random splits; not accounting for temporal relationships.

[7] VulnPatchPairs; built using vulnerable functions from CodeXGLUE/Devign and their corresponding patched versions.

[8] Fu and Tantithamthavorn [20] provide a version with a random fixed test/train split.

[9] Contains both real-word and synthetic data.

[10] Constructed by manually reviewing a subset of data from VUDENC [64], Big-Vul [92], and CrossVul [99].

[11] Constructed by merging data from Big-Vul [92], CrossVul [99], CVEfixes [88], and DiverseVul [90].

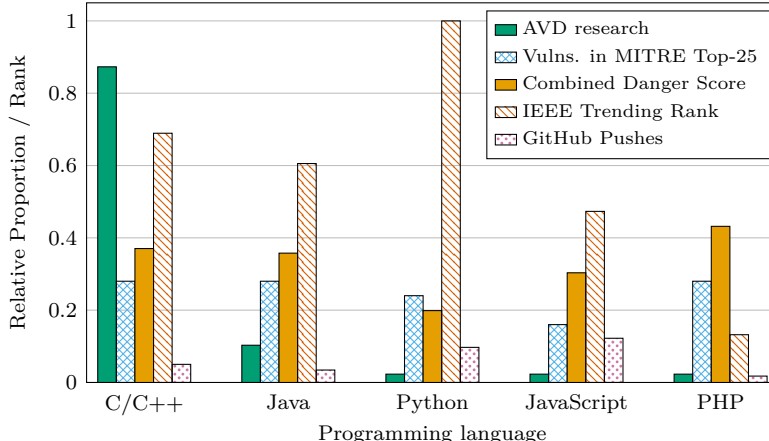

Figure 12: Commonly-used programming languages according to (1) their proportion in included AVD research, and (2) their proportion of the 2023 top 25 MITRE dangerous software weaknesses they are vulnerable to, (3) the proportional score of the MITRE danger score, (4) their 2024 IEEE Trending rank (proportion of the most-used language, Python), and (5) their proportion in GitHub pushes from Q1 of 2024. Metrics collected to reflect the period the included research was published.

vulnerable language, followed by C/C++, Java, and JavaScript. Yet, PHP currently makes up less than 10% of AVD research, disproportionate to its relative vulnerability.

To capture the prevalence of a language, we use the 2024 IEEE Spectrum Top Programming Languages ranking [112]. Languages are ranked as a proportion of the most popular language, Python, with C/C++ ranked second and Java third. Despite this, Python is only represented in less than 10% of AVD research. Another metric for prevalence is the relative proportion of *GitHub pushes* from Q1 2024. According to this metric, JavaScript is the most prevalent, followed by Python and then C/C++. This analysis is supported by Li et al. [155] who report that, among 7,113 popular open-source projects, C, C++, Java, JavaScript, and Python account for approximately 22%, 14.5%, 13.5%, 8%, and 7%, respectively, when measured by code size.

This analysis uncovers clear discrepancies between AVD research and real-world use and needs. Although the literature frequently claims that ML4AVD solutions proposed for C/C++ would simply transfer to other languages, this claim is not sufficiently backed by evidence. Programming languages can differ both syntactically and semantically, and different languages often suffer from different security vulnerabilities [166].

**Recommendations.** To better reflect the diversity of real-world languages and vulnerabilities, future research should diversify its focus to include languages such as Python and JavaScript, due to their significant popularity, along with Java and PHP due to their susceptibility to vulnerabilities. Furthermore, we suggest that future work should propose ML4AVD solutions targeting more complex multi-lingual software projects, as a recent survey of professional developers found that a typical software project has an average of 7 different programming languages [159]. As with previous pain points, datasets should support this by providing training data across multiple languages.

Tooling no longer imposes as significant a limitation on which languages can be supported. For example, Joern language support has extended to include Python, PHP, Java, JavaScript, and other languages [145]. Thus, any ML4AVD that integrates graph-based representations is no longer limited to C/C++.

> 🗄 **Pain Point 5: Widely-used datasets are unrepresentative**
>
> Datasets do not represent a realistic distribution of labels, projects, and vulnerabilities, and some widely-used datasets include synthetic samples.

**Description and Motivations.** A related issue to Pain Point 4 is the wider unrepresentative nature of prolific datasets. The most widely-used dataset in the analyzed papers is Devign [84] (or the CodeXGLUE [97] derivative), which has an unrealistic balanced proportion of vulnerable and benign labels. This is explicitly mentioned as a motivation issue for producing the imbalanced ReVeal dataset [9]. When using imbalanced datasets, some included articles used under-sampling of benign samples (e.g., [16, 53, 59, 74]), over-sampling

of vulnerable samples using SMOTE [115] (e.g., [5, 9, 13, 80]) or both (e.g., [32, 79]) to reduce the imbalance in the training data. In ablation studies, re-balancing was shown to have a positive effect on $F_1$-score [5, 9].

Both the Devign and ReVeal datasets include samples from only two software projects each, which limits the diversity of code present in the dataset, and the types of vulnerabilities. Similarly, the dataset used by Harer et al. [22] is sourced from multiple projects but Debian is the source for 63% of the samples. The VulDeePecker dataset [33] only includes two vulnerability types, which motivates the introduction of the similar SySeVR dataset [31] containing 126 vulnerabilities, though this is only a fraction still of the over 900 types listed on CWE [166]. Some widely-used datasets (e.g., [33, 49]) also include significant proportions of synthetic samples from SARD [95] alongside samples collected from real-world data.

More recently, datasets have been created that only contain paired samples of vulnerable code and the corresponding patched code [17, 96, 100], which are inherently balanced. These datasets are used, in part, to demonstrate problems with the field, which we discuss in Pain Point 7, Pain Point 9, and Pain Point 11.

**Implications.** Real-world vulnerability distributions are heavily skewed, with the largest proportion of files being benign. Training deep learning methods on datasets that over-represent vulnerabilities may bias models, and training only on datasets containing vulnerable-patched sample pairs excludes features present in files that are not security-relevant. Both of these biases may produce more false positives [108]. Conversely, the proportion of vulnerable samples in a truly realistic dataset may be too low to efficiently train models [9, 90] and biases models towards benign predictions. The proportion of vulnerable samples in the testing and validation sets also significantly affects evaluation methodology and reported results, which we discuss in Pain Point 10 and Pain Point 11, and the ratio of a re-balanced dataset has been found to influence the trade-off between precision and recall [9]. Synthetic samples do not provide a realistic depiction of the complexity of real-world vulnerabilities and, thus, may not provide useful signal for training ML4AVD models. With LLM-based methods, synthetic data also poses an additional challenge when it is generated by AI, as it can lead to model collapse [127]. Low variety in vulnerability types and source projects reduces models' usefulness and generalizability to unseen code [143]. Additionally, even when sampling from multiple projects, if one project dominates the dataset, it may bias models toward the vulnerabilities and code patterns in that repository, although project-balanced batch sampling does not appear to improve results [143]. These limitations on datasets can result in models that overfit the training data and fail to generalize to unseen data, as discussed further in Pain Point 9.

**Recommendations.** We encourage future work that produces representative datasets sampling from real-world data while integrating the recommendations from the subsequent Pain Point 6, Pain Point 7, and Pain Point 8. We reiterate the recommendation from Pain Point 3 that any future work consuming existing datasets should critically evaluate them.

To handle imbalanced data for training, we recommend conservatively under-sampling benign samples rather than over-sampling vulnerable samples to avoid introducing unrealistic synthetic samples. When under-sampling, the corresponding patched sample for each vulnerable sample should be retained within the training set to provide fine-grained contrastive signals. Semantics-preserving data augmentation can be used for over-sampling vulnerable samples, but it can cause overfitting on code introduced by the augmentation [179]. Furthermore, the training distribution can also be tuned empirically: in the adjacent imbalanced classification domain of malware classification, tuning algorithms have been proposed to find the training class ratio that maximizes a target performance metric on a held-out validation set [148, 172]. However, the resulting training set should still reflect that benign samples are more common; a perfectly-balanced training set is not desirable as it can bias models towards vulnerable predictions and ignore features from benign files that are not part of a vulnerable/patched pair. In such imbalanced settings, re-weighting the loss function inversely proportional to the label ratio has also been empirically shown to improve performance [90].

Validation and testing sets must be kept heavily imbalanced to ensure accurate calibration and evaluation on realistic data. Representative datasets can help prevent overfitting, which is discussed further in Pain Point 9, and allow for more realistic evaluations, which we discuss further in Pain Point 10. The high imbalance inherent to realistic datasets has a profound impact on metrics, which we cover in Pain Point 11.

🗄 **Pain Point 6: Widely-used datasets suffer from quality issues**
Commonly-used datasets include high proportions of mislabeled samples and duplicate data.

**Description and Motivations.**  Dataset quality is a well-known problem in the ML4AVD field [9, 119, 138, 180, 102] and poor-quality ground truth labels are cited as a motivation for choosing synthetic datasets (e.g., [70]). In a manual review, Croft, Babar, and Kholoosi found that a subset of the Big-Vul [92] dataset contained 45% mislabeled samples; Devign [84] contained 20%, and D2A [104] contained 71% mislabeled samples. This is partially attributed to the collection techniques used, which take security-related commits and label functions that have been changed as vulnerable pre-commit, even if the change is unrelated or not directly security-related. Additionally, datasets contained significant duplicate samples and had inconsistent labels for some duplicate entries, e.g, as two security-related changes to a single function would be processed to label the intermediate version of the function once as non-vulnerable and subsequently as vulnerable. These results are backed by subsequent analysis by Ding et al. [91]. Similarly, Risse and Böhme [180] found high-proportions of false positive labels in random subsets of DiverseVul [90] (35%), Devign [84] (50%), and Big-Vul [92] (61%). Conversely, some datasets that aim to produce high-quality, low-noise data, tend to be smaller and cover a low number of vulnerabilities (e.g., SVEN [93]). The quality of available datasets is mentioned as a threat to validity in some included articles (e.g., [7, 21, 62, 69, 79, 85]). To overcome this issue, some works use manual checks (e.g., [7, 68]) or introduce new datasets or data collection methods (e.g., [9, 54, 60, 68, 74]).

**Implications.**  Low quality data produces less capable ML4AVD models which are affected by biases in the data, learning spurious features [108] (cf. Pain Point 9). Poor quality datasets also make evaluating models' true capabilities difficult, as incorrect labels in the evaluation set will impact the reported metrics. This is further discussed in Pain Point 10. Additionally, duplicate samples within the training data introduces biases towards those samples and negatively affect the performance of the resulting models [106]. When duplicate samples occur between between training and test data, it undermines the validity of the evaluation. This is one form of test dataset leakage, which is discussed in wider detail in Pain Point 8. When removing duplicates model performance decreases compared to reported results [12, 119, 91].

**Recommendations.**  Any future work should be aware of the impact of using datasets suffering from high levels of noisy labels or duplicate data, as identified by the literature [9, 119, 91, 180]. If using these datasets, pre-processing steps may help remove some duplicate and mislabeled data, though any pre-processing should be clearly described, and any refined datasets created as part of the work should be made available in the service of fairly comparing work (cf. Pain Point 10). Pre-processing steps, such as slicing, may also introduce duplication [9, 80], so de-duplication may be required in such cases even when the underlying dataset has unique samples. We recommend critically assessing datasets and using those that resolve the limitations we identify; existing widely-used datasets with known quality issues should be used only for evaluation to facilitate comparison with previous solutions. As no dataset is without limitations, future work should transparently evaluate and discuss the effect of their chosen dataset on the results [108].

Manually labeled datasets, such as SVEN [93], have higher sample quality than older datasets based on collecting vulnerability related commits, but are smaller and unrepresentative (cf. Pain Point 5). Newer datasets have been constructed specifically in response to the quality issues of the older datasets. The PrimeVul dataset [91] is curated from existing datasets to increase the quality of the labels. The PairVul [17] and ReposVul [102] datasets, which are both created from vulnerability fixing commits, undergo further filtering based on changes to the targeted files to avoid irrelevant and duplicate data being included. The CleanVul dataset [96], which uses an LLM-based commit message filtering system, offers multiple levels of confidence, with a trade-off between label quality and dataset size.

> 🗄 **Pain Point 7: Datasets lack important information**
>
> Widely-used datasets do not contain CWE type, do not explicitly pair related vulnerable and fixed samples, do not provide meta-information about the vulnerability (CVE, bug reports, discussions).

**Description and Motivations.** Beyond lacking important code-level context as discussed in Pain Point 2, datasets lack meta-information required for ML4AVD. The widely-used Devign dataset [84] and its derivatives do not include information about the vulnerability type. Many datasets used in the analyzed work (e.g., [31, 33, 49, 84]) do not make explicit the link between vulnerable samples and their patched equivalents, which makes them unsuitable for training solutions that specifically look to use paired samples to learn only vulnerability-specific features and reduce overfitting [54]. Only Big-Vul [92] includes information about the vulnerability in the form of a CVE ID, although the newer datasets in Table 6 are more likely to include CVE ID data. No dataset includes other meta-information surrounding the vulnerability, such as bug tracker issues, mailing lists, or merge request discussions for the patch.

**Implications.** The lack of CWE ID contributes to the focus on binary formulations described in Pain Point 1. Lacking explicit links between vulnerable/patched sample pairs prevents leveraging pairs during training to increase separation between classes, which leads to spurious correlations. It also prevents evaluations from verifying that models can indeed separate the vulnerable and patched versions of the same code, which has been identified as a limitation of existing ML4AVD solutions [179], including LLM-based solutions [91, 17, 127, 198]. We discuss spurious correlation and overfitting further in Pain Point 9 and the metrics on paired samples that can be used to identify these issues in Pain Point 11. As LLM-based solutions are being increasingly used, which can make use of additional non-code information [17, 73], the lack of meta-data and vulnerability-related discussion in the datasets impedes this avenue of development.

**Recommendations.** Datasets should at least include all information that a human expert may use to identify a vulnerability, which allows novel methods to integrate code and non-code data (commit messages, bug tracker issues, mailing lists, or pull/merge request discussions). As most datasets include at least repository and commit information, it is possible for future work that uses datasets without explicit pairs to reconstruct the link between vulnerable and patched samples, then use them for training and evaluation. As commit messages may reference bug trackers or other sources of discussion, these non-code sources may also be available for some projects for automated collection. Additional non-code information may also help models explain the causes of a vulnerability and the mechanics that led to its classification, which feeds into the wider explainability discussion in Pain Point 12. The VulZoo dataset [101], which enriches vulnerabilities with additional information, including mailing list discussions, can be used to source non-code data, which may especially benefit LLM fine-tuning.

> 🗄 **Pain Point 8: Datasets leak test data**
>
> Many of the widely used datasets lack explicit train/test splits, and where it is present, assignment is random, ignoring temporal relations between code samples in the same project; duplicate samples and data snooping by pre-trained large language models aggravate these issues.

**Description and Motivations.** The majority of datasets used in the included articles do not provide fixed training/validation/test subset splits. This is recognized as an issue and fixed splits of existing datasets have been made available. The defect component of CodeXGLUE [97] is a fixed split of the *Devign* dataset. Fu and Tantithamthavorn [20] publish a split of the Big-Vul [92] dataset, which is used by subsequent works [48, 53]. Where a fixed split is present in datasets used by the analyzed work, assignment is randomized [20, 97, 49, 104]; when using datasets without a fixed split, analyzed articles most commonly use random assignment (e.g., [6–9, 12, 14, 19, 24, 38, 59, 62, 66, 73, 81, 84]), which is motivated by mirroring previous work and enabling better baseline comparisons [20, 48, 59, 66] and rarely use date-based splits (e.g., [65]). Even with fixed test sets, datasets that are made public in their entirety are likely to have their test data included in the training data from LLMs [17]. Additionally, synthetic samples from SARD [95], sampled by some datasets

(e.g., [33]), include the label ("bad") in the variable names, which requires pre-processing to remove [70] (see also Pain Point 5).

**Implications.** The lack of fixed train/test splits complicates comparing multiple ML4AVD solutions, as reported results may depend on the composition of the splits [89] (cf. Pain Point 10). More troubling, samples collected from a project are likely not to be entirely independent and later samples may provide information about earlier samples. Therefore, randomly splitting datasets into test and training sets is likely to leak test set information to the ML4AVD models, allowing models to 'time travel', which inflates reported results. When a time-based split is used instead of a random split, model performance is shown to decrease [103]. Outside of ML4AVD, temporally consistent splits have become common practice in the adjacent domain of malware detection [116, 147, 148, 160–162, 172, 203] and, more widely across ML for cybersecurity, the necessity to capture the non-stationarity inherent in cybersecurity tasks has been raised [108, 163]. Additionally, as discussed in Pain Point 6, many datasets suffer from duplicate samples. Splitting duplicates between the training and test sets allows models to memorize samples, again inflating reported results [9]. Pre-trained LLMs introduce further challenges, as their training data incorporates large quantities of online resources, LLMs may implicitly include test data in their training [125, 127] and measures to detect test data leakage, such as canary strings, have been integrated into benchmarks [191]. Agentic LLM-based solutions, which have been shown to exhibit reward hacking [200], may access openly available test data.

**Recommendations.** Future work should use datasets derived from real-world data with fixed train/test splits, especially recent datasets [89, 91, 103] that explicitly account for temporal correlation between samples and split the test data chronologically. This prevents test set leakage and provides a more accurate evaluation, while using datasets that filter out duplicates prevents leakage caused by duplicate samples being split between train and test sets (cf. Pain Point 6). Some datasets provide a public leaderboard [97, 104]. To minimize data leakage and memorization issues, especially with LLM-based solutions, we recommend future datasets retain a hidden test set for use with a public leaderboard [114]. This strategy is used in landmark benchmarks (e.g., [174]) to reduce the risk of test data leakage and preserve the benchmark's usefulness over time. It is particularly important for LLM-based solutions, where a model may have been trained on code that is included in future benchmarks, even if the benchmark's release date is after the knowledge-cutoff date reported in the model card.

> 🎓 **Pain Point 9: Solutions suffer from overfitting and poor generalization**
>
> Poor training practices lead to models learning spurious correlations and, subsequently, poor generalization to unseen source code.

**Description and Motivations.** ML4AVD solutions are known to overfit training data and learn spurious features. In the analyzed articles, overfitting is seen as a potential threat to validity (e.g., [16]). Therefore, some of the analyzed works attempt to reduce the possibility of overfitting through generic deep learning methods, such as reducing the number of parameters [16, 19, 27, 53], using dropout (e.g. [4, 19, 36, 39, 41, 50, 62, 64, 75, 79]), early termination/low number of epochs [21, 26, 77], and other techniques [5, 12, 32, 36, 38, 49, 61, 68, 84]. This aligns with the common view of AVD as purely an ML classification task, as discussed in Pain Point 1. Only a minority of the included papers uses ablation studies to analyze how different aspects of the ML4AVD solutions and validate that each component is contributing to any claimed improvements over the state-of-the-art [4, 5, 7, 9, 11, 13, 17, 18, 20, 30, 35, 37, 40, 42, 47, 48, 53, 57, 59, 61–63, 65, 67–69, 72, 73, 76, 82].

Overfitting is sometimes discussed when comparing work with baseline results, especially in explaining inconsistent values compared to those originally reported [9, 38, 54, 80, 85] (cf. Pain Point 10). Evaluating baselines on different datasets/projects than those used for training produces worse results [24], which is indicative of overfitting to training data. Some included articles use cross-dataset/project evaluation for this reason [13, 53]. Some approaches to mitigating overfitting focus on the limitations of the dataset. The ratio of vulnerable and benign samples is a consideration for overfitting [1, 86] and some works pre-process datasets to balance vulnerable and benign samples [86] (cf. Pain Point 5) and to remove duplicates [12], with the explicit

goal of reducing overfitting. In general, test set leakage (cf. Pain Point 6) constitutes a significant cause of overfitting [9], including for LLM-based solutions, which appear to exhibit better performance on data before the training cutoff date [73]. Overfitting is also seen as a result of the small number of vulnerable examples in datasets [54], with some works adopting data augmentation techniques in response [37, 48]. Additionally, two identified works try to combat overfitting by training explicitly on vulnerable/patched sample pairs [54, 55], which requires that this data be present in the dataset, as mentioned in Pain Point 7. Spurious features at the text-level affect both deep learning-based methods [53] and LLMs [17]. Finally, explainability techniques have been used to identify spurious correlations [7, 9] (cf. Pain Point 12).

**Implications.** Overfitting produces artificially inflated results that do not hold up in a real-world setting, as it produces models that are unable to generalize to unseen data. Much of the analyzed work uses some generic techniques to reduce overfitting, but, as previously noted, a minority apply domain-specific techniques. These techniques have also been used in the related literature to investigate overfitting and spurious correlations across ML4AVD. Data augmentation through semantics-preserving code mutations is used to demonstrate that existing techniques overfit to specific syntax and text tokens, rather than semantic features [89, 179, 100], with a large emphasis on memory allocation code [108, 89]. Overfitting and poor generalization are also demonstrated through cross-project [90] and cross-augmentation [179, 100] evaluations. Evaluation on paired vulnerable/patched samples shows that ML4AVD solutions, including LLM-based solutions, are unable to consistently distinguish them correctly [91, 17, 198]. The lack of code-level context in function-level datasets (cf. Pain Point 2 and Pain Point 7) may contribute to models learning spurious features, as samples frequently lack enough information to learn the correct features of vulnerable code [180].

**Recommendations.** Beyond applying the standard techniques that seek to reduce overfitting, future work should look at the specific problems that cause vulnerability detection solutions to overfit and fail to generalize to novel projects. As a fundamental step, we recommend using representative, high-quality datasets that contain multiple granularities and context for each sample, as per the discussions in Pain Point 2, Pain Point 5, and Pain Point 6. Having a fixed test set and preventing test set leakage can reduce the overfitting to the test set (cf. Pain Point 8). Normalizing code representation and other semantics-preserving code transformations may help with generalization [143], although the effect appears to be limited [179]. Additionally, contrasting paired vulnerable and patched samples, which is supported by datasets that make such pairs explicit (cf. Pain Point 7), may help models learn the specific differences between them, reducing spurious correlations. Evaluating on paired samples can subsequently be used to determine if an ML4AVD solution is able to truly distinguish vulnerable and patched code [91, 17] (cf. Pain Point 10). We discuss the metrics that support this in Pain Point 11. Evaluating on different projects, datasets, or with different data augmentations than the training set can provide better insights into how well the model generalizes to unseen code (cf. Pain Point 10). Ablation studies can validate the chosen architecture and that no extraneous complexity exists in the system. Explainability and interpretability techniques, which we cover further in Pain Point 12, beyond their immediate goals, may also help identify spurious correlations in model output.

> ⚖️ **Pain Point 10: Evaluation makes it difficult to measure progress and applicability**
>
> Comparability across works is problematic, due to evaluation methodology, limitations of datasets, and difficulty in reproducing baseline results; evaluation does not give a sense of realistic performance.

**Description and Motivations.** The poor reproducibility of results is noted early in the period we analyzed, with Chakraborty et al. [9] not being able to reproduce the results of Zhou et al. [84]. Zhou et al. did not make their implementation public, which required it to be re-implemented, and the public Devign dataset [84] contains code sourced from two open source projects (FFmpeg, Qemu), while the model was trained using a broader dataset containing samples from four projects (additionally, Linux and Wireshark). This example illustrates some of the wider issues and the interplay between datasets, open science, and reproducibility. Similar issues with reproducibility are reported by other analyzed articles [47, 51, 81]. It is common (38/87) for code and other artifacts (e.g.,model weights/checkpoints) not to be made available in

the analyzed work [1, 2, 4, 6, 13–16, 18, 22, 24, 28, 33, 34, 37–41, 44, 49, 51, 52, 54, 56–58, 69, 70, 72, 75, 77, 79, 82, 84–87]. In at least one case, the code was made available but is no longer public [47]. Similarly, many works construct datasets that are not subsequently made available [1, 4, 10, 14, 16, 18, 22, 24, 37, 41, 44, 52, 57, 72, 75, 79, 82, 86] and the D2A [104] is no longer available through IBM, who originally produced it, but only as mirrors. Pre-processing steps are another factor that may influence results, as the pre-processing step may not be fully described and the data that results is not made available. As a positive counterexample, Fu and Tantithamthavorn [20] produced a version of the Big-Vul dataset [92] with fixed splits, which was used in later work specifically to help comparability. As mentioned in Pain Point 7, many widely-used datasets lack fixed train/test splits which is recognized as a barrier to reproducibility [65], as result may be affected by the allocation of samples, especially when duplicate samples and temporal correlation are present (cf. Pain Point 6 and Pain Point 7). Finally, any use of closed-weight models through an API produces reproducibility issues, as models may change, even without public announcements. For this reason, Yang et al. [73] use open weight models in their work. Although a majority of analyzed articles consider the realism of the setting in some form, this is mainly limited to the composition of the datasets; some analyzed articles performed user studies to determine real-world applicability of their work (e.g., [7, 17, 32]).

**Implications.** Without effective evaluations, progress in the field of ML4AVD is difficult to measure. As seen in Figure 1, which we used as a motivating example, the field of ML4AVD does not appear to report consistent improvements, though individual articles try to demonstrate improvement over their chosen baselines. ML4AVD is a real-world problem and the analyzed ML4AVD research presents a gap as a result of the problem formulation (Pain Point 1) and granularity. Datasets lacking code context (Pain Point 2) and having unrepresentative distribution of languages (Pain Point 4), vulnerabilities and vulnerability types, and sampled projects (Pain Point 5) further undermine the real-world applicability of the analyzed works. Because of this, evaluations do not convey how well the ML4AVD solutions can be applied in real-world settings. Additionally, poor evaluation hides the problems of overfitting and poor generalization to unseen data, covered in Pain Point 9, and further undermines the practical utility of the solutions.

**Recommendations.** Evaluation should aim to validate that the proposed ML4AVD solution is an improvement over the state-of-the-art, which requires easily available and reproducible work. Therefore, future work should open-source all code, including any pre-processing steps, and model weights and checkpoints to aid reproducibility and future baselines. Furthermore, it should fully describe hyper-parameters, including setting and publishing random seed values. When using LLM-based methods, prompts should be considered a constituent part of the parametrization of the model. When using open-weight models, the exact version should be specified; for closed-weight models, this becomes more difficult, but where possible, model names should be fully specified (e.g., timestamps for OpenAI models) and API parameters published. Thus, future work should provide as close to a complete replication package as possible. Furthermore, the versions of any datasets, tools, and other resources used for training or any replication packages used for establishing baselines should be clearly specified. Any datasets collected as part of future work or any refinements to existing datasets should, licensing permitting, be made public. These should include the recommendations discussed in Pain Point 5, 6, 7, and 8. As datasets have been known to become unavailable when hosted independently [208], we strongly recommend using dedicated hosting platforms that support versioning for all open-sourced artifacts. Available platforms include Zenodo[1], OSF[2], Hugging Face[3] (which supports leaderboards with hidden test sets [114] to avoid Pain Point 8), and for code, GitHub[4], GitLab[5], or Codeberg[6]. Subsequent work can easily acquire the resources required to reproduce results. As different components of an ML4AVD solution may have different impacts on the results, evaluation against baselines should try to capture how components may improve the baseline solutions, which is especially important with any pre-processing techniques. This complements ablation studies, as discussed in Pain Point 9, and provides a more rigorous evaluation of each component of the ML4AVD solution.

---

[1] https://zenodo.org/
[2] https://osf.io/
[3] https://huggingface.co
[4] https://github.com
[5] https://gitlab.com
[6] https://codeberg.org/

Additionally, evaluation methodology should ensure any ML4AVD solution is able to generalize and that it has real-world applicability, as opposed to 'lab-only' evaluation [108]. Therefore, evaluation should occur on representative imbalanced real-world datasets (cf. Pain Point 5) with realistic time-based train/test splits (cf. Pain Point 7). Additionally, integrating methods designed to detect overfitting (cf. Pain Point 9) into evaluation methodologies can help a model's generalizability. At a minimum, evaluation should measure whether a model can distinguish samples in vulnerable/patched pairs either as a subset of samples in the test set or as a separate evaluation step using balanced datasets containing such paired samples [91, 17, 100]. Metrics are a key component of evaluation, including paired samples; we discuss these separately in Pain Point 11. For new agentic LLM-based solutions, evaluation should ensure that results are valid, and agents are not reward-hacking or extracting correct answers from available data [200]. Additionally, user studies with practitioners, as undertaken by some of the included papers, may help provide further insights into applicability.

> ⚖️ **Pain Point 11: Metrics do not capture realistic performance**
>
> The majority of ML4AVD works use only classification metrics, without accounting for class imbalance or detecting overfitting.

**Description and Motivations.** As noted in subsection 4.6, the most common metrics used in the literature are classification metrics, which aligns with the observations in Pain Point 1. Frequently, the motivation for the choice of metrics, particularly $F_1$-score, *Accuracy*, *Precision*, *Recall*, and *ROC-AUC*, is the task being (binary) classification (e.g., [1, 4, 9, 10, 19, 20, 26, 41, 45, 57, 73, 74]); however, more commonly the choice of metrics is motivated as being 'standard', 'popular', or 'widely-used' (e.g., [3, 6, 7, 9–12, 14, 16, 18, 23–25, 31–34, 39, 40, 42, 47, 54, 62, 65, 67–70, 76, 77, 80–82]) or with reference to specific prior work (e.g., [13, 30, 42, 47, 59, 66, 69, 71, 73, 81]). In some works that reference prior work, consideration is specifically given to comparability when choosing metrics [13, 21, 73] (cf. Pain Point 10). No explicit motivation for the choice of metrics is given in a significant proportion of work (e.g, [2, 8, 15, 28, 29, 35, 37, 46, 49–53, 55, 195, 58, 60, 61, 63, 72, 75, 78, 79, 84, 86, 87])

A small proportion of works dive deeper into the interactions between metrics, real-world vulnerability detection, and datasets. Consideration is given to the impact of *FP* and *FN* in the context of VD [3, 33, 57], with *FP* reducing trust and causing alert fatigue, and *FN* allowing vulnerabilities to slip through. Precision and recall are seen as having a trade-off relationship [18, 27, 49], which causes most works to use $F_1$ to try to incorporate the two metrics into a single value, though one work uses $F_2$ [27], giving higher weight to precision, and one work [83] uses $F_{0.5}$, giving higher weight to recall, as each cites different errors as having a higher impact in VD. As noted, *PR-AUC* is used infrequently, though it is motivated in one work as better for imbalanced datasets [23]. Dataset imbalance affects the choice of metrics in a minority of included articles [5, 12, 14, 17, 18, 21–23, 31, 32, 38, 39, 43, 48, 53, 57, 59, 74, 83, 85], leading some works to diminish the importance of or to exclude accuracy [14, 18, 39, 64], or to reweigh metrics according to the label ratio [17, 21, 43, 59]. Unbiased metrics are also used to handle imbalanced data, with MCC being used more [21, 31, 38, 74, 85] than Informedness and Markdness [12, 13]. Paired accuracy, the number of paired vulnerable/patched samples that are both classified correctly, is used in one work [17] to detect spurious correlations (cf. Pain Point 9).

**Implications.** Imbalanced data is a widely-studied issue in machine learning [146]. It strongly affects the ML4AVD field due to the inherent imbalance of vulnerabilities. To understand how models may perform in real-world settings, evaluating on realistic, imbalanced test sets is paramount (cf. Pain Point 10), and this requires metrics that can capture performance in imbalanced datasets. If not accounted for, this leads to the base rate fallacy, which causes accuracy to overestimate performance [108]. This greatly affects included articles that only report *Accuracy* (e.g. [2, 46, 50, 86]), as a model that only classifies code as benign will have high accuracy on a realistic, highly imbalanced dataset. Similarly, a conservative model may have high *Precision* on an imbalanced dataset but very low *Recall*. The trade-off between *Precision* and *Recall* is closely tied to the effects of false positives and false negatives on ML4AVD, which can depend on the specific program. Due to the label imbalance, even apparently low *FPR* can be misleading, as it may still

lead to high absolute numbers of false alerts, which cause alert fatigue. Additionally, due to its dependence on *FPR*, *ROC-AUC* can similarly be misleading. In turn, the cost of false negatives is highly dependent on the application and the time at which the ML4AVD solution is being run; for example, missing a new vulnerability introduced in a code change compared to missing an existing vulnerability while scanning extant code. Due to this, different projects may have different inherent utility functions for an ML4AVD solution and different tolerances for each error type. Thus, the $F_1$-score, which equally weighs *Precision* and *Recall*, does not capture the trade-off well, as it provides information at a specific threshold [91]. In contrast, the *PR-AUC* curve offers a more comprehensive, threshold-independent characterization of the trade-off between *Precision* and *Recall*, enabling practitioners to select operating points aligned with their specific utility functions. Additionally, the use of only classification metrics does not account for the specific challenges of ML4AVD, and only one analyzed article identifies this as a threat to validity [62]. None of the traditional classification metrics capture whether models can distinguish between vulnerable and patched versions of the same sample.

**Recommendations.** We recommend that future work evaluates on heavily imbalanced realistic test sets to establish real-world performance and report the *PR-AUC* curve to make the trade-off between *Precision* and *Recall* explicit in a threshold-independent manner. As even low *FPR* can incur substantial practical costs, we argue that *Recall* at a given *FPR* (denoted as *Recall@FPR*) provides a more actionable measure of effectiveness. We therefore encourage researchers to report *Recall@FPR* alongside widely used metrics such as the F1 score. To support comprehensive evaluation, *Recall@FPR* should be presented across a range of FPR values. For continuity with existing work, standard summaries such as the $F_1$-score can be provided alongside these metrics. Additionally, to ensure any metrics can be computed when the work is replicated or used as a baseline, future work should provide a multi-class confusion matrix as part of the replication package (cf. Pain Point 10).

When evaluating how well an ML4AVD solution can differentiate the vulnerable and patched samples in a pair, future work should report the proportion of pairs for which both samples were classified correctly [91, 17], which gives an insight into whether the model captures relevant features of vulnerable samples or learning spurious features. As with the main results, providing a full confusion matrix for the paired samples can better inform reproduction of the work and its use as a baseline.

Although publishing new CVEs has been used as a metric for real-world applicability (cf. Pain Point 10), it incentivizes spurious vulnerability reporting [184]. Therefore, we do not recommend using it as a metric in future work.

---

⚖️ **Pain Point 12: Solutions exhibit poor explainability and interpretability**

> Explainability and interpretability are not a consideration for much of the included articles, providing only a label without providing a motivation for why the model arrived at the classification.

---

**Description and Motivations.** Although literature on interpretability and explainability has tried to separate the terminology – explainability meaning a post-hoc process, while interpretability is inherent to the process [209] – the analyzed literature uses these terms interchangeably, even within the same article. As the analyzed articles focus on deep learning models (including language models), the internal mechanisms of models do not provide interpretability. This is recognized as a challenge in the included articles (e.g., [19]), though it is often left as 'future work' [19, 21, 32, 36, 54, 55, 59, 60, 76, 85]. The explainability of graph neural networks is used to justify the choice of ML approach in some included articles [66, 79] (cf. subsection 4.5) and Li, Wang, and Nguyen [29] use subgraph-based explainability to provide finer output granularity (cf. Pain Point 2). Cao et al. [7] specifically set out to create an explainable GNN-based ML4AVD solution. Similarly, attention-based explainability is also used to produce finer-grained output [20, 26]. Articles that performed user studies evaluated the explainability of the models [7, 32], which indicates that it is an important consideration for real-world applicability.

**Implications.** Explainability and interpretability provide information linking vulnerability classification with potential causes (cf. Pain Point 1). This can feed into further steps of the vulnerability life cycle, helping practitioners and improving real-world applicability. The complexity of deep learning and large language models makes them inherently un-interpretable, therefore research into explainability techniques has been an active area. Post-hoc explanations for GNN-based ML4AVD solutions have been proposed [117, 131, 142, 193] and have been integrated into one included article to provide better output [29]. Beyond their use in refining output granularity, explainability techniques have been used to reveal overfitting in ML4AVD solutions [108, 89] (cf. Pain Point 9).

As language models have become more prevalent in the ML4AVD space, they have been seen as a potential solution for explainability, but *self-explanation* may not be *faithful* to those processes [105, 123, 197], producing plausible but misleading explanations. This has been observed in LLM-based ML4AVD applications as well [188, 198], with correct categorization paired with incorrect explanations and vice-versa.

**Recommendations.** A key component of explainability is implicitly resolved by following the recommendations in Pain Point 1 and Pain Point 2,as more precise and contextually informed outputs from ML4AVD systems enhance the explainability of classification results. Generic black-box explainability tools(e.g, [137, 157, 177]) may provide some benefit [108]. For GNN-based solutions, specific explainability solutions have been developed [7, 131, 158, 204]. Research using LLM-based methods should go beyond self-explanation, look at the ever-evolving field of LLM explainability [206], and apply mechanistic interpretability techniques [110, 189], although such techniques are only applicable to open-weight models and not closed-weight models.

---

**Overall Trends**

The limitations of ML4AVD we identified are strongly correlated with each other, showing a strong interplay between the different facets of machine learning (formulation, datasets, training, and evaluation) which can be seen in Figure 11. The datasets in Table 6 show that, while there have been improvements, none of the identified datasets bring together all the recommendations we put forward. Across many of the pain points we identified, we note a strong trend of following previous research among the analyzed articles, which we have already filtered to identify influential, highly-cited work.

**What is the Direction for Detection?**

We recommend that future work critically engages with previous work in the field, not just at the technical level, but also conceptually. Thus, future work should seek to answer the answer the correct question, taking into account the very real-world problem of vulnerability discovery in the context of the vulnerability life cycle. We recommend future work to critically evaluate the datasets it uses, according to the criteria we lay out, and choose datasets that are useful in answering their research question, rather than allowing dataset availability to influence the direction of the research.

---

## 6 ML4AVD in the Age of Agents

There have been two major changes in recent years that have the potential to shape the future of the AVD field: first, the AI Cyber Challenge (AIxCC) [120], one of the largest initiatives to date aimed at making automated detection, exploitation, and repair of vulnerabilities practical through advancements in LLMs; second, the rise of agentic AI, which has demonstrated rapidly improving capabilities in both producing and analyzing code at scale. Having identified the pain points of ML4AVD research in section 5, we now examine how the field is evolving with respect to these two factors. Section 6.1 uses AIxCC as a case study to assess the extent to which its design choices align with the pain points and recommendations we have raised. Section 6.2 then reflects on whether dedicated ML4AVD research is still needed when general-purpose agents are becoming increasingly capable at analyzing code and finding vulnerabilities.

## 6.1 Case Study: Was AIxCC a Step in the Right Direction?

Organized by DARPA as a successor to the 2016 Cyber Grand Challenge (CGC) [122], AIxCC, which held the semi-finals in 2024 and the finals in 2025, represents the largest and most recent effort to bridge the gap between ML4AVD research and operational software security. We treat it here as a case study: a lens through which to evaluate whether the competitive framing and design choices of AIxCC move the field closer to the practical requirements identified throughout this paper.

### 6.1.1 AIxCC Overview

AIxCC was a competition designed to advance the state of autonomous vulnerability detection, exploitation, and repair. Teams developed Cyber Reasoning Systems (CRSs) that operated on version-controlled repositories containing full source code in C/C++ or Java of real-world open-source projects, including the Linux kernel, nginx, and Jenkins. The vulnerabilities were synthetically introduced by the competition organizers into the commit history of these host repositories, and participants were required to identify the vulnerability-inducing commits. To support detection, AIxCC provided public and private test suites alongside *sanitizer*-based validation harnesses that confirmed whether a vulnerability had been triggered. A perfect solution had to autonomously perform detection, exploitation, and patching (removing the vulnerability while preserving the functionality of the code). A distinguishing feature of AIxCC was its emphasis on the use of commercial large language models. While AIxCC did not mandate a specific approach, LLM usage was heavily encouraged, with substantial LLM credits (from OpenAI, Anthropic, Google) made available to participating teams. Notably, the CRSs operated with restricted compute and internet access; external LLM use was limited to an approved list of commercial models. However, open-source models could be deployed locally. This setup allowed the competition to assess how a mix of frontier commercial LLMs and team-specific tooling can augment the vulnerability remediation process. Submissions were evaluated across five areas of excellence: handling of large codebases, support for multiple programming languages, coverage of diverse vulnerability types, accuracy of vulnerability discovery, and effectiveness of patching.

### 6.1.2 🎯 Research question

AIxCC required submissions to provide information about the detected vulnerability type by selecting the specific sanitizer expected to be triggered by the vulnerability. Since each sanitizer corresponds to a class of vulnerability, AIxCC used a formulation that is adjacent to the CWE-ID formulation, but without explicitly requiring the CWE-ID. Furthermore, AIxCC adopted a repository-level input granularity and a commit-level output (detection) granularity, with multiple vulnerabilities potentially present per repository.

**Pain Point 1.** AIxCC partially addresses the dominance of binary classification. By requiring sanitizer selection, AIxCC mandates a form of vulnerability type-specific labeling, which is a step toward a more realistic formulation than pure binary classification. Although the detection component of the formulation still centers on classification rather than explicitly producing richer context (e.g., triggering conditions or code paths) for downstream vulnerability life cycle stages, AIxCC's broader pipeline does require an exploit and patch for a complete score, which implicitly rewards solutions that provide rich, relevant information for the patching step.

**Pain Point 2.** AIxCC's separation of input and output granularities follows our recommendation, allowing approaches to leverage a wider context while still providing localized detection output. That being said, commit-level detection may still be too coarse-grained, especially for larger commits that affect multiple functions or files, and falls short of the recommendation to keep output granularity as fine-grained as feasible.

**Pain Point 3.** By defining its own evaluation structure rather than relying on existing benchmark datasets, AIxCC avoids the feedback loop where available datasets shape problem formulations. This independence from existing datasets allows the competition to set its own expectations for what a solution should produce, rather than inheriting the limitations of prior benchmarks.

### 6.1.3 </> Programming language

AIxCC solutions had to handle vulnerabilities in both C/C++ (e.g., Linux kernel and nginx) and Java (e.g., Jenkins) repositories, requiring that CRSs be generic enough to operate across multiple languages.

**Pain Point 4.** This multilingual requirement addresses the observation that ML4AVD research overwhelmingly targets C/C++ alone. Furthermore, the use of full source-code repositories, rather than pre-processed intermediate representations, avoids tool-driven language limitations, where the availability of pre-processing tools (e.g., Joern for C/C++ CPGs) implicitly constrains language support. Although AIxCC considered only two language families, supporting both C/C++ and Java already forces systems beyond narrowly language-specific pipelines and may encourage more genuinely multilingual ML4AVD approaches.

### 6.1.4 🛢 Dataset

Before the AIxCC semi-finals, the organizers published three challenge projects for participants to test their CRSs. The semi-finals used five scored challenge projects, of which *nginx* (containing 14 vulnerabilities) has since been released and used in recent ML4AVD work, such as [181]. The finals were substantially larger, with 53 challenge projects derived from 24 open-source repositories and containing 63 synthetic vulnerabilities. Following the finals, DARPA has begun progressively releasing competition challenges, alongside the competition infrastructure, telemetry, and harnesses, via the AIxCC Competition Archive [121].

**Pain Point 5.** The use of real-world codebases as host projects provides a more realistic evaluation context than fully synthetic datasets, such as those drawn from SARD. However, the vulnerabilities themselves were synthetically introduced into the commits, which limits the dataset's utility for studying the detection of emergent real-world vulnerabilities.

**Pain Point 6.** The use of expert-crafted vulnerabilities with sanitizer-based validation harnesses provides a degree of label quality assurance that is absent from many existing datasets, which suffer from noisy or incorrect labels. However, DARPA reports that teams also discovered 18 real, non-synthetic vulnerabilities during the finals, in addition to the seeded synthetic vulnerabilities. This is encouraging from an operational perspective, but it also exposes a residual dataset-quality issue: in realistic host projects, the ground truth may be incomplete, as CRS findings may correspond to genuine but unlabeled vulnerabilities.

**Pain Point 7.** AIxCC does not readily provide vulnerable-patched pairs of software, but these can be reconstructed from the commit history by reverting vulnerability-inducing commits or applying the ground-truth patches. The version-controlled repository format also inherently provides richer non-code context (e.g., commit messages, patch history) compared to the isolated function-level samples that dominate existing datasets.

**Pain Point 8.** During the competition itself, the scored challenge projects were withheld from the competing teams, ensuring a fair comparison free from information leakage. In this respect, AIxCC stand in contrast to the majority of existing AVD datasets, which lack both fixed train/test splits and never had a publicly withheld evaluation phase. However, where the introduced vulnerabilities where re-additions of previously disclosed CVEs, the commercial LLMs used in the challenge may have already encountered the original disclosures during training, leading to potential test data leakage. Furthermore, as DARPA progressively releases the challenges, the released data may end up in the training corpora of future LLMs, reintroducing the test data leakage that the competition structure was designed to avoid.

### 6.1.5 🎓 Approach

Although AIxCC did not explicitly require teams to develop CRSs using a specific approach (either ML-based or not), the application of LLMs as part of the CRS was heavily encouraged. The focus on commercial LLMs is well-motivated by their success in related domains and their ability to produce natural-language outputs that, while not always faithful (cf. Pain Point 12), are easier for downstream practitioners to consume than opaque numeric scores.

**Pain Point 9.** AIxCC's structure does not directly target the overfitting and spurious-correlation concerns raised in Section 5, but it does test generalization implicitly: scored challenge projects were withheld during

the competition, differed from the public exemplars, and spanned multiple repositories, so a CRS overfitting to any single project could not carry that performance forward.

### 6.1.6 ⚖️ Metrics and Evaluation

The AIxCC scoring algorithm combines four assessed metrics to compute the overall team score: the Vulnerability Discovery Score (VDS), the Program Repair Score (PRS), a Diversity Multiplier (DM), and an Accuracy Multiplier (AM). The VDS and PRS are combined via a weighted logarithmic function, where patching is weighted approximately three times more heavily than discovery alone. The DM rewards CRSs that discover and patch vulnerabilities across a broad range of CWE classes, indirectly encouraging language diversity since certain CWEs cluster differently by language. The AM penalizes teams for inaccurate or rejected submissions, computed as a ratio of accurate to total submissions, thereby discouraging strategies that rely on submitting large numbers of invalid results.

**Pain Point 10.** AIxCC provided a standardized evaluation environment with consistent infrastructure, test sets withheld during the competition, and a unified scoring metric. For the scope of the competition, this addresses the problem of poor comparability, where self-reported results on varying dataset splits with differing baselines hinder meaningful comparison across approaches. However, the competition's combined scoring metric and evaluation environment are not directly transferable to standard ML4AVD evaluation, however, the release of competition infrastructure may enable partial reuse in future work.

**Pain Point 11.** The multipliers reward solutions that have low rates of false positives and can detect a diverse set of vulnerabilities across projects and languages, both of which directly address concerns raised in Section 5: the AM penalizes the false-positive-driven alert fatigue noted in our metrics discussion, and the DM actualizes the call for richer, type-aware detection from Pain Point 1. The combined score therefore functions as an application-aware utility function rather than a generic classification metric, aligning with the recommendation that ML4AVD evaluation reflect the costs and benefits of deployment.

**Pain Point 12.** AIxCC required not only detection but also exploitation and patching, which implicitly demands reasoning about vulnerability semantics rather than relying on superficial correlations. Producing a working exploit or a correct patch requires a degree of understanding beyond what binary classification can capture. Although the commercial LLMs used in the challenge can produce natural language explanations, as mentioned in Section 5, such self-explanations are not always faithful. Furthermore, the competition did not explicitly evaluate the quality of explanations, leaving this as an area for future improvement.

### 6.1.7 Summary

Overall, AIxCC represents a significant step forward in aligning research efforts with real-world requirements. Its most impactful contributions are: the decoupling of input and detection granularity (Pain Point 2); the evaluation of multiple languages (Pain Point 4); the in-competition enforcement of withheld test sets (Pain Point 8); the validation of labels through sanitizer-based verification (Pain Point 6); the use of an application-aware scoring function rather than generic classification metrics (Pain Point 11); and, the embedding of detection within a full vulnerability life cycle pipeline , which addresses the broader concern that AVD is too often treated as a standalone classification task (Pain Point 1). However, AIxCC does not fully resolve several key Pain Points: synthetic injection of disclosed CVEs may have leaked into LLM training (Pain Point 8); commit-level output remains coarser than recommended (Pain Point 2); and explanation quality is not directly evaluated (Pain Point 12). We contend that AIxCC is best understood not as a solution to the pain points of ML4AVD, but as a proof of concept that the research community can build upon: one that demonstrates how design can reshape research incentives toward more practical and operationally relevant approaches. For a complementary analysis focused on the specific CRS design choices made by competing teams, we refer the reader to the recent SoK on AIxCC [205].

### 6.2 Is ML4AVD still needed?

The rise of capable agentic LLMs raises a fair question for the future of the field: if general-purpose code-capable agents can already locate vulnerabilities in real codebases, as AIxCC's CRSs demonstrated, then

is dedicated ML4AVD research still needed or will it be subsumed by progress in foundation models? We argue that it remains both relevant and necessary, for three reasons.

**Pain Points are Approach-Agnostic.** The methodological issues surfaced in section 5 are not artifacts of any single model architecture. They persist across the dominant approaches we observed (CNNs, RNNs, GNNs, and LLMs) and, in several cases, are *aggravated* by the move to LLMs: opaque training corpora make data contamination harder to rule out (Pain Point 8), agentic systems are vulnerable to reward hacking when test data is reachable [200], and LLM self-explanations are known to be unfaithful (Pain Point 12). Future work, regardless of the approach chosen, benefits from confronting these pain points directly rather than rediscovering them under a new architectural label.

**Specialized Models Complement Agents.** Agentic detection pipelines are increasingly capable of integrating heterogeneous signals through tool use, which positions specialized ML4AVD models as components within broader systems rather than as standalone end-to-end solutions. Hybrid architectures already exemplify this: combining LLMs with GNNs over graph-based IRs [58] captures long-range dependencies that flat token sequences struggle with, and agents can orchestrate symbolic execution [151], concolic testing [187], and guided fuzzing [164] alongside ML components. Furthermore, specialization remains valuable beyond raw capability: it supports local deployment (privacy), low-latency in-IDE inference (cost), and reproducible evaluation through fixed open weights (Pain Point 10).

**Need for Open Research.** Even as frontier models (e.g., Anthropic Mythos [111]) push the commercial state-of-the-art on vulnerability discovery, their closed nature limits reproducibility (Pain Point 10), interpretability (Pain Point 12), and applicability in air-gapped or privacy critical settings, which make sending code to external API infeasible. Furthermore, closed-source models are subject to opaque changes by their providers, which can silently affect performance in ways that consumers cannot anticipate or control [107]. Therefore, there is still scope for smaller open-weight models as they are needed both for scientific research and understanding as well as deployment scenarios where external large commercial models are not applicable.

Finally, as agentic coding frameworks are accelerating the rate at which software and vulnerabilities are produced, we argue that the demand for scalable and accurate detection grows as well. Therefore, what has changed is not whether AVD research is needed, but what shape it should take: less function-level binary classification on saturated benchmarks, more focused attention on building principled foundations, as discussed in this survey.

# 7 Related work

The challenges in the automated vulnerability detection space have been examined by a number of works. This survey complements and integrates the work in the ML4AVD space that identifies limitations and pain points. Table 7 shows a selection of related papers and the limitations and issues they identify. These provide valuable insights and some empirically demonstrate limitations with ML4AVD solutions; unfortunately, all the works we identified cover only parts of the overall picture and do not thoroughly identify causal connections between the limitations. This survey is influenced in its presentation by the work of Arp et al. [108], who look at a wider set of works, including ML4AVD, but also other problem domains. Harzevili et al.[139] surveyed 67 papers using ML- and DL-based AVD approaches until 2022. Chakraborty et al. [9] and Chakraborty et al. [113] find overfitting and poor real-world performance in ML4AVD works. Bi et al. [109] focuses solely on the evaluation and benchmarking aspects of AVD, reviewing existing datasets, and discussing efforts to create better benchmarking procedures. Dataset quality is discussed by a number of works [119, 138, 180]. Risse and Böhme [180] tackle the topic of granularity in the AVD domain and ask whether the widely used function-level granularity is sufficient in practice. Risse and Böhme [100] and Ding et al. [91] demonstrate overfitting and propose evaluation on pairs of vulnerable and patched samples. Sheng et al. [188] survey LLM use in vulnerability detection and identify a subset of the pain points. Similarly, Evertz et al. [127] focus on the specific "pitfalls" of LLM-based AVD solutions. Kaniewski et al. [149] include a subset of limitations in the ML4AVD space, but conflate input and output granularities in their recommendations. Zhou et al. [207] survey 58 papers that use LLMs for vulnerability discovery and repair, and discuss a subset of the limitation we identify in the context of using LLMs for AVD. Shimmi, Okhravi,

Table 7: Related works on automated software vulnerability detection and the limitation identified

| Cat. | | Pain point / Year (20xx) / Articles | 149 ’25 263 | 100 ’24 6 | 108 ’21 30 | 139 ’23 67 | 91 ’24 n\s | 180 ’25 81 | 127 ’25 72 | 9 ’22 4 | 113 ’24 4 | 109 ’23 n\s | 188 ’25 60 | 133 ’25 208 | 190 ’25 98 | 207 ’24 82 | Ours ’26 87 |
|---|---|---|---|---|---|---|---|---|---|---|---|---|---|---|---|---|---|
| Formulation | 1 | Classification problem | ✗ | ✗ | ✗ | ✗ | ✗ | ✓ | ✗ | ✗ | ✗ | ✓ | ✓ | ✗ | ✓ | ✗ | ✓ |
| | | Binary classification | ✓ | ✗ | ✗ | ✗ | ✗ | ✓ | ✗ | ✗ | ✓ | ✓ | ✓ | ✗ | ✓ | ✗ | ✓ |
| | 2 | Function-level input | ✓† | ✗ | ✗ | ✗ | ✗ | ✓ | ✗ | ✗ | ✓ | ✓ | ✓ | ✗ | ✗ | ✓ | ✓ |
| | | Function-level detection | ✓† | ✗ | ✗ | ✗ | ✗ | ✓ | ✗ | ✗ | ✗ | ✓ | ✗ | ✗ | ✓ | ✗ | ✓ |
| | 3 | Shaped by dataset | ✗ | ✗ | ✗ | ✗ | ✗ | ✗ | ✗ | ✓ | ✓ | ✗ | ✗ | ✗ | ✗ | ✗ | ✓ |
| Lang. | 4 | Focus on C/C++ | ✓ | ✗ | ✗ | ✗ | ✗ | ✗ | ✗ | ✗ | ✗ | ✗ | ✓ | ✓ | ✓ | ✗ | ✓ |
| | | Tools determine language | ✗ | ✗ | ✗ | ✗ | ✗ | ✗ | ✗ | ✗ | ✗ | ✗ | ✗ | ✗ | ✗ | ✗ | ✓ |
| | 5 | Unrealistic label ratio | ✓ | ✓ | ✓ | ✗ | ✓ | ✗ | ✗ | ✓ | ✓ | ✓ | ✓ | ✗ | ✓ | ✗ | ✓ |
| | | Synthetic-only | ✓ | ✗ | ✓ | ✗ | ✓ | ✗ | ✓ | ✓ | ✓ | ✓ | ✗ | ✓ | ✓ | ✓ | ✓ |
| Dataset | 6 | Low-quality labels | ✓ | ✓ | ✓ | ✓ | ✓ | ✓ | ✓ | ✗ | ✓ | ✓ | ✓ | ✓ | ✓ | ✓ | ✓ |
| | | Duplicate data | ✓ | ✗ | ✗ | ✗ | ✓ | ✓ | ✗ | ✓ | ✗ | ✗ | ✓ | ✓ | ✓ | ✗ | ✓ |
| | 7 | No patched code | ✗ | ✓ | ✗ | ✗ | ✓ | ✗ | ✗ | ✗ | ✗ | ✗ | ✗ | ✗ | ✗ | ✗ | ✓ |
| | | Lacking non-code data | ✓ | ✗ | ✗ | ✗ | ✗ | ✗ | ✓ | ✗ | ✗ | ✗ | ✓ | ✗ | ✗ | ✗ | ✓ |
| | 8 | Lacking test/train split | ✓ | ✗ | ✓ | ✗ | ✓ | ✗ | ✗ | ✗ | ✗ | ✗ | ✗ | ✓ | ✗ | ✗ | ✓ |
| | | Test data leakage | ✓ | ✗ | ✓ | ✗ | ✓ | ✗ | ✓ | ✗ | ✗ | ✗ | ✓ | ✓ | ✗ | ✗ | ✓ |
| Training | 9 | Spurious correlation | ✓ | ✗ | ✓ | ✗ | ✓ | ✓ | ✓ | ✓ | ✗ | ✗ | ✗ | ✗ | ✗ | ✗ | ✓ |
| | | Not on patched pairs | ✗ | ✓ | ✗ | ✗ | ✓ | ✗ | ✗ | ✗ | ✗ | ✗ | ✗ | ✗ | ✗ | ✗ | ✓ |
| | | Overfit/poor generalization | ✓ | ✓ | ✓ | ✗ | ✗ | ✓ | ✗ | ✓ | ✓ | ✗ | ✗ | ✓ | ✗ | ✗ | ✓ |
| Evaluation | 10 | Inconsistent baselines | ✓ | ✗ | ✓ | ✗ | ✗ | ✗ | ✓ | ✓ | ✗ | ✓ | ✗ | ✓ | ✓ | ✗ | ✓ |
| | | Poor practical evaluation | ✓ | ✗ | ✓ | ✗ | ✓ | ✗ | ✗ | ✓ | ✓ | ✓ | ✓ | ✗ | ✗ | ✓ | ✓ |
| | 11 | Classific. metrics only | ✗ | ✗ | ✓* | ✗ | ✓ | ✗ | ✗ | ✗ | ✗ | ✗ | ✗ | ✗ | ✗ | ✗ | ✓ |
| | | Label imbalance | ✓ | ✗ | ✓ | ✗ | ✓ | ✗ | ✗ | ✗ | ✗ | ✗ | ✗ | ✗ | ✗ | ✗ | ✓ |
| | 12 | Poor explainability | ✓ | ✗ | ✗ | ✓ | ✗ | ✗ | ✗ | ✗ | ✗ | ✓ | ✓ | ✓ | ✓ | ✗ | ✓ |

† Discusses input and output granularity but conflates the two, suggesting that future work should "explore coarser detection granularity".

* Discusses the use of inappropriate metrics in general terms, rather than the specifics of vulnerability detection.

and Rahimi [190] survey 98 articles published between 2018 and 2023, providing an in-depth taxonomy of the research, and discuss some of the limitations, though less thoroughly. Germano et al. [133] survey LLM use for detection and repair, including a broader discussion on explainability in LLMs.

We recommend to readers that are interested in more specific areas of the ML4AVD space to consider this work complementary to many of the works mentioned above, especially those that provide empirical evidence for the pain points (e.g., [113, 91, 100]) or insights on specific areas or approaches (e.g., datasets [109, 119] or LLMs [127]).

# 8 Threats to validity

Despite our rigorous methodology in collecting and systematizing research, several potential limitations may impact our findings: (1) Our survey focuses exclusively on techniques aimed at detecting security vulnerabilities, although methods targeting general bug and defect detection may provide additional insights. This scope was chosen to ensure a thorough and relevant analysis specific to AVD. (2) Our search terms,

while necessarily specific to manage the volume of papers, may have inadvertently excluded some relevant studies. To mitigate this, we supplemented our search with a manual review, as outlined in subsection 3.1, to capture critical work that could be missed by our initial search. (3) Our methodology to determine high-impact papers is citation-dependent which may be skewed, while some related work focuses on top-level venues; we chose this methodology as publication at top-level venues does not guarantee the work will be influential in the field. (4) We focus on static analysis in service of the vulnerability discovery component of the life cycle; although dynamic analysis techniques, such as fuzzing, show an overlap with the exploitation stage of the vulnerability life cycle, we may have excluded relevant ML4AVD work. (5) The time period required for work to be cited excludes recent papers with high impact; we partially mitigate this by analyzing recent datasets, and including recent surveys and empirical studies into the discussion and recommendations. (6) The time period means LLMs are under-represented compared to their increase in their usage in recent work; although LLMs have unique pain points that future work should be aware of, much of the discussion is agnostic of the ML approach and LLM-specific pain points are touched in the wider discussion. (7) Our discussion and recommendations are based on the included articles and related literature, but subjectivity may influence our perception of the pain points of ML4AVD and their severity.

# 9 Conclusion

The analysis we performed on 87 influential ML4AVD works across six dimensions – problem formulation, granularity, programming language, datasets, evaluation metrics, and ML approach – and identified twelve interconnected pain points that explain why the field has failed to show consistent progress despite a decade of sustained effort. Our analysis indicates self-reinforcing cycles are present in ML4AVD research, which requires future work to critically evaluate what came before, rather than look at incremental improvements. The rise of agentic LLMs does not diminish the need for dedicated ML4AVD research. The methodological pain points identified here are approach-agnostic and, in several cases, are aggravated by LLMs. The need for improved open, small, and efficient ML4AVD models is still present, even if proprietary models are becoming increasingly capable. The direction for detection is clear; what remains is the will to follow it.

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

## A    Performance Meta-analysis

This appendix expands on Figure 1, the motivating figure presented in section 1, by examining the self-reported performance of ML4AVD solutions across the three most-used datasets. We consider the three most-used AVD datasets: Devign [84], ReVeal [9], and Big-Vul [92]. Of the 87 articles reviewed in our survey, 19 reported the $F_1$-score on at least one of the aforementioned datasets.

In Figure 1, each point represents an ML4AVD solution's publication year on the x-axis, and the corresponding $F_1$-score reported on the corresponding dataset on the y-axis. We chose the $F_1$-score because (1) it is the most prevalent metric in the literature, as evidenced by Figure 10, and (2) it provides a more holistic view than *Accuracy*, *Precision*, or *Recall* in isolation. We note, however, that $F_1$ is itself a limited metric for ML4AVD, as discussed in Pain Point 11. The analysis below is, therefore, best read as a meta-analysis on what is reported in the literature and not to determine which approach is best.

For the **Devign** dataset, the figure does not show a clear improvement over time and surprisingly the highest scoring solution was Devign [84], which is the article that originally introduced the dataset. As discussed in Pain Point 10, subsequent work [9] was unable to reproduce these results. Apart from Devign itself, the top-performers on the Devign dataset are: (1) *VulGDA* [28] ($F_1 = 67.7$), a cross-domain solution leveraging GGNN and domain adaptation (DA) [128], where DA is a branch of transfer learning involving pre-training on one dataset (source domain), and later fine-tuning on task-specific data (target domain), and (2) *AMPLE* [67] ($F_1 = 66.9$), which trains an edge-aware GCN using the AST and CFG. Both of these solutions leverage GNN-based approaches.

For the **ReVeal** dataset, focusing on the top-performing solutions each year reveals a slight improvement up to 2023. The top performers are: (1) *VulGDA* [28] ($F_1 = 71.4$), the top-performing solution for the *Devign* dataset above, and (2) *CPVD* [79] ($F_1 = 70.2$), which is another cross-domain solution using GAT and DA. Both top-scoring solutions combine GNNs with DA, suggesting that transfer learning across projects is particularly effective for this dataset. A pattern consistent with the recommendation in Pain Point 9 to evaluate across, rather than within, projects.

The performance on the **Big-Vul** dataset is more variable than that of Devign and ReVeal, with two outlier solutions reporting exceptionally good performance: (1) *DeepDFA+UniXcoder* [53] ($F_1 = 96.5$), a GNN-based solution aiming to simulate data flow analysis combined with the LLM UniXcoder [136], and (2) *LineVul* [20] ($F_1 = 91$), a pre-trained CodeBERT LLM, later fine-tuned for vulnerability detection. Similar to Devign and ReVeal, the top-performers on the Big-Vul datasets use very similar approaches: a pre-trained LLM adapted to AVD through (1) fine-tuning [20], or (2) by concatenating its output features with those generated by a GNN [53]. The credibility of these high $F_1$-scores is, however, difficult to establish given the data leakage concerns raised in Pain Point 8: pre-trained LLMs may have encountered Big-Vul

test samples during their training, and Big-Vul is also among the datasets reported to contain a substantial proportion of mislabeled samples [119, 91](Pain Point 6).

Despite the lack of a clear overall trend through the years and the self-reported nature of the results, a pattern emerges across the three datasets: *cross-domain ML4AVD solutions consistently appear among the top performers.* These solutions leverage the paradigm of transfer-learning by first learning a base pre-trained model from a widely-varied heterogeneous dataset and then fine-tuning on the target dataset. One plausible explanation is that cross-domain training reduces overfitting to dataset-specific spurious correlations (Pain Point 9).

Furthermore, the data demonstrates the frequent introduction of new solutions that fail to surpass the per-dataset top performers identified above, indicating a lack of fair comparisons to state-of-the-art baselines. This reinforces the recommendation in Pain Point 10 that future ML4AVD work explicitly benchmark against the top-performing solutions for each dataset used, to (1) ensure the proposed solution matches or exceeds state-of-the-art performance, and (2) surface the reproducibility issues affecting many of the results for top-performing solutions, as illustrated by the Devign case above.

