# OpenReview forum: "Direction for Detection: A Survey of Automated Vulnerability Detection and all of its Pain Points"
_TMLR — Under review for TMLR_

### Review · Reviewer_ntEx · 2026-06-19

**Summary Of Contributions:**

This is a systematization-of-knowledge survey of machine learning for automated vulnerability detection (ML4AVD). The authors build a corpus through a documented, multi-stage screening pipeline. They start from 965 collected articles, de-duplicate and topic-screen down to 395 candidate solution papers plus 76 empirical studies, apply an age-adjusted citation-frequency threshold (at least five citations per year), and finish with 87 highly cited works after full-text screening. They then characterize this corpus across six dimensions: problem formulation, input and output granularity, target programming language, datasets, evaluation metrics, and ML approach. The quantitative picture is clear and well documented. The field concentrates heavily on binary classification (86 percent of articles), on C/C++ (87 percent of single-language articles), and on function-level input and output (63 percent).

The core contribution is the synthesis layer on top of this systematization. The authors define twelve "pain points" that span the pipeline and argue that these are not independent problems but a set of mutually reinforcing loops, where function-level datasets encourage function-level solutions that motivate more function-level datasets, and where the scarcity of high-quality benchmarks makes self-reported performance an unreliable measure of progress. Each pain point is paired with concrete recommendations. The paper then applies its own framework as a lens on AIxCC, the 2024 to 2025 DARPA AI Cyber Challenge, scoring how well that effort addresses each pain point, and closes with a reflection on whether dedicated ML4AVD research is still needed in an era of capable agentic models.

I think this is a strong and useful paper, and my recommendation is acceptance after a round of revisions. The points below explain what I see as its value, where the evidence and the claims are not yet fully aligned, and what would make it more durable as a reference.

What I see as the real value of this work:

The most valuable thing here is the integrative diagnosis. A reader who already knows two or three of these pain points in isolation will still get something new from seeing them laid out as a connected system, with the feedback loops made explicit and tied back to a measured corpus. The field has needed a single reference that says, in one place and with data, why a decade of effort has not produced a clear upward trend. This paper does that.

The systematization itself is a reusable asset. The six-dimension breakdown, the dataset comparison in Table 6, and the related-work comparison in Table 7 are the kind of tables that other researchers will cite and build on for years. Table 6 in particular, which maps datasets against the specific attributes that matter for the pain points, is a genuine reference contribution.

The AIxCC case study is timely and well executed. Rather than treating the competition as a success story or a failure, the authors use it as a structured test of their own recommendations, and they are even-handed about what it does and does not resolve. That section turns the survey from a backward-looking catalog into something forward-looking and actionable, which is exactly what a good systematization should do.

Finally, the methodology is transparent and the threats-to-validity section is honest. The screening funnel is reproducible, the citation threshold is stated and justified, and the authors are candid about the limits of a citation-based corpus and the under-representation of very recent LLM work.


Key strengths:

A clear, reproducible corpus-construction methodology with explicit inclusion and exclusion criteria.
A well-documented quantitative systematization across six dimensions, with the distributions that back the headline claims.
A novel and useful integrative framing that connects previously isolated limitations into a coherent set of reinforcing loops.
High-value reference tables (datasets, related-work coverage) and an even-handed, timely AIxCC case study.
Honest treatment of scope and limitations.


Key weaknesses (expanded below):

The headline word "causal" is stronger than the evidence, which is qualitative synthesis rather than a demonstration of directionality.
The motivating Figure 1 relies on self-reported F1 across heterogeneous datasets and splits, which the paper itself argues elsewhere is unreliable.
Several external factual claims in the AIxCC section should be checked against primary sources.
A number of duplicate-word typos and one research-question inconsistency need a copyedit pass, which matters more than usual for a reference work.

**Additional Comments:**

This is a recommendation to accept after revisions, not a rejection. I think the systematization and the integrative framing are a real contribution, and the dataset and related-work tables will be reused by others.
To make the discussion phase efficient, the two changes that would most affect my assessment are (1) aligning the "causal" framing with the evidence, either by softening it to a mutually reinforcing framing or by adding a temporal analysis that shows directionality within the corpus, and (2) reframing the motivating Figure 1 honestly, since it relies on the same self-reported numbers the paper later argues are unreliable. The AIxCC finals figures should also be checked against primary DARPA sources.
The remaining items, including the copyedit pass and the single-versus-two research-question inconsistency, are minor and easy to address.

**Audience:**

Yes

**Audience Explanation:**

The audience is clear and substantial. Researchers in ML for security, in program analysis, and in benchmark and dataset design will all find direct value here, and the systematization tables give them something concrete to reuse. Beyond the immediate ML4AVD community, the paper speaks to a broader TMLR readership interested in how benchmark and dataset choices can quietly steer a whole subfield toward an artificial version of a real problem, which is a pattern that recurs well outside vulnerability detection. The AIxCC case study and the discussion of whether specialized detection models still matter alongside capable agentic models add timeliness that will draw readers who are tracking how foundation models are reshaping applied security research. This criterion is clearly met.

**Broader Impact Concerns:**

The paper does not contain a Broader Impact statement, and given that it is a survey of methods rather than a new capability, none is strictly required. I would, however, encourage the authors to add a short dual-use paragraph, because the subject matter invites it. The paper itself documents that increasingly capable models, including the agentic systems in the AIxCC case study and the Mythos model it cites, are now discovering large numbers of real vulnerabilities in open-source software. A survey that lays out how to build better detection systems and better benchmarks also, by the same token, lays out how to build better discovery tools, and improved public benchmarks can aid offensive use as readily as defensive use. A brief, measured note on responsible disclosure norms and on the offense-defense balance of better ML4AVD would fit the work well and would be appropriate for a reference paper in this area. This is a recommendation rather than a blocking concern.

**Claims And Evidence:**

Yes

**Claims Explanation:**

I want to be clear about how I am reading TMLR's first criterion for a survey. The relevant questions are whether the corpus and its quantitative characterization are accurate, whether the represented literature is faithfully described, and whether the synthesis is sound and properly hedged. On those terms the paper largely succeeds. The distributional claims are directly backed by the corpus and the tables, the descriptions of individual works that I am familiar with are accurate, and the related-work positioning in Table 7 is fair. My "yes" is conditional on a set of fixable issues rather than on any new experiment.

The one claim that currently outruns its evidence is the central framing that the pain points are "causally inter-meshed" through "feedback loops." The paper supports the existence of each pain point well, and it supports the co-occurrence of the dominant choices (binary, C/C++, function-level) with strong numbers. What it does not provide is evidence of direction or strength for the causal arrows in Figure 11. The argument is a plausible and well-told narrative, but it is narrative. For a survey this is acceptable as long as the language matches the evidence, so I am asking the authors either to soften "causal" to something like "mutually reinforcing" or "co-evolving," or to add what evidence they can for directionality, for example a temporal analysis showing that dataset releases precede and predict shifts in formulation choices in the corpus. This is the single most important revision, because the causal framing is the paper's main differentiator from prior surveys.

A related tension is the motivating Figure 1, which plots self-reported F1 across three datasets and concludes there is "no clear upward trend." Pain Point 10 then argues at length that self-reported numbers across differing splits and baselines are not comparable. The paper is using as its opening evidence exactly the kind of measurement it later tells the reader to distrust. I do not think this sinks the claim, since the honest reading is that even optimistically biased self-reported numbers fail to show progress, but the figure should be framed that way explicitly, the selection and normalization of points should be described clearly (Appendix A is referenced for this and should carry the detail), and the "no upward trend" statement would be stronger as a simple trend test rather than visual inspection.

On factual accuracy in the AIxCC case study, I spot-checked the externally verifiable claims. The reference to Anthropic's Mythos model is legitimate and correctly characterized. The AIxCC finals figures are close to what public reporting describes, but the synthetic-vulnerability count should be verified, since secondary sources vary on the exact number and the paper states 63 where some reporting indicates a different figure. Because a case study's authority depends on these numbers, I would like to see them cited to primary DARPA sources.

None of these issues concern the integrity of the corpus or the soundness of the systematization, which are the heart of the paper. They are about aligning a few strong words with the evidence, framing the motivating figure honestly, and pinning down external facts. With those addressed, the claims are well supported.

**Requested Changes:**

I have separated changes that I consider important for acceptance from those that would strengthen the work. I have tried to make each one specific.

Important for acceptance:

Align the "causal" framing with the evidence. Either soften the central claim from "causally inter-meshed" to a correlational or mutually reinforcing framing throughout the abstract, introduction, and Section 5, or add concrete support for directionality. A temporal analysis within the corpus, showing for example that dataset releases precede and predict subsequent formulation and granularity choices, would let you keep stronger language. As written, Figure 11 presents directed arrows that the data do not yet justify.

Reframe and document the motivating Figure 1. State explicitly that the figure uses self-reported, non-comparable numbers and that the point is that even these optimistically biased numbers show no progress. Move the selection and normalization details for the plotted points into the referenced appendix and make them complete enough to reproduce. Consider replacing "no clear upward trend" with the result of an explicit trend test.

Verify the AIxCC figures against primary sources. Please confirm the finals numbers, in particular the count of synthetic vulnerabilities, against primary DARPA or competition sources, and cite them. Public secondary reporting varies on these figures, and the case study's credibility depends on getting them right.

Fix the research-question inconsistency. The paper defines a single "RQ" but then refers to "RQ1" and "RQ2" when describing how the sections map to the analysis. Decide whether there is one research question or two, and make the section roadmap consistent. The roadmap in the introduction also lists Section 7 and Section 9 but omits Section 8 (Threats to validity).

Run a careful copyedit pass. For a reference work, small errors undercut authority more than usual. Examples to fix include the duplicated "machine learning machine learning" in the abstract and introduction, "are propagate widely" in the introduction, "between between training" in Pain Point 6, and "the answer the answer the correct question" in the Overall Trends box. A full pass would likely catch more.


Would strengthen the work:


Show sensitivity to the citation threshold. The corpus depends on the choice of at least five citations per year. A short robustness check, showing that the six-dimension distributions are stable if the threshold is moved up or down, would reassure readers that the headline percentages are not artifacts of one cutoff.

Justify the scope exclusions briefly. The exclusion of smart-contract vulnerability detection is notable, since it is a large ML-for-vulnerability subfield, and the same is true for malware and binary-only work. A sentence or two on why these are out of scope and how their exclusion might bias the picture would help.

Reassure on coverage of empirical studies. You separate out 76 empirical studies and fold them into the discussion rather than the citation-filtered corpus. A brief statement confirming that the well-known large empirical studies of deep learning for vulnerability detection are represented in that discussion would close a potential gap a reader might wonder about.

Strengthen Figure 11. If you keep a relationship diagram, consider adding edge direction or relative strength, or a short formal description of what each edge means, so the figure carries more than an illustrative role.

Make the recommendations easier to act on. A summary table mapping each pain point to its recommendation and to the datasets or benchmarks in Table 6 that already satisfy it would turn the advice into a checklist that future authors can follow directly. Much of this information is already present and could be consolidated.

---

> ### Author Response · Authors · 2026-07-09
> **Reply to reviewer ntEx**
>
> Thank you for your thorough, constructive feedback.
>
> > The field has needed a single reference that says, in one place and with data, why a decade of effort has not produced a clear upward trend. This paper does that.
>
> That was our intent with this work and we are happy it came across in your reading of the survey.
>
> ## Important
>
> ### Claim of "causally inter-meshed" through "feedback loops."
>
> We understand this concern and we can change the wording to "mutually reinforcing" or "co-evolving", so it doesn't seem like we are overselling the analysis.
>
> ### Figure 1
>
> Yes, the idea of presenting the figure early is exactly that "even optimistically biased self-reported numbers fail to show progress", so we will adopt the proposed changes to make it clearer that is the intent.
>
> ### AIxCC
>
> We used the AIxCC SoK by Zhang et al [205] to source the reported numbers and we will clarify this by citing it inline with the reported numbers. Furthermore, we will double-check the reported numbers against primary sources (https://archive.aicyberchallenge.com/repoviz/) and add citations to this where appropriate.
>
> ### Others
>
> * The reference to multiple research questions is fixed. This was left from a previous draft.
> * Section 8 (Threats to validity) will be listed in the roadmap
> * We have fixed the copyedit issues highlighted and we did a pass to fix other small issues.
>
> ## Strengthen
>
> ### Sensitivity
>
> We set the threshold of 5 citations per year as this is the mean citation count for computer science papers 1 year after publication. We can expand on the reasoning for using citation-based instead of venue-based filtering with references to meta-studies on systematic reviews to strengthen this part of the methodology.
>
> ### Exclusion
>
> Smart contract vulnerabilities were excluded as the literature on vulnerabilities in smart contracts considered vulnerability types that are out of scope for other software: logic errors, blockchain timing (frontrunning), and exploitation via market manipulation, which would bias our analysis. Although there is significant overlap, we believe smart contracts would benefit from a bespoke analysis, separate from other software.
>
> The reason for excluding malware is that, although it is a software-based detection task, the objective is different as security vulnerabilities are more localised and generally assumed to be mistakes in writing software, while malware classification is inherently adversarial and focused on the intent of the application as a whole. Malware detection is a sister-field, so many of the techniques and limitations overlap but only in part.
>
> We did not explicitly exclude binary-only work and we do include binary work such as [51], [26], [72], [75].
>
> ### Empirical studies
>
> We will add an early explicit mention of the influential empirical studies that we included in our analysis, such as [108], [180]. We already list some of them in the related work comparison table.
>
> ### Figure 11
>
> We will adjust this figure to make it more explicit as a summary of Section 5, by adding arrows to show the direction of the relation and label/explain the relationships better. The information we are trying to distill in this diagram is dense and complex so it's difficult to strike a good balance between information-dense and still useful/not overwhelming to a reader.
>
>
> ### Summary table of recommendations
>
> We can distill the recommendations into an accessible checklist for future researchers to refer to as they conduct their own work. An AVD preflight check.
>
> ## Broader Impact statement
>
> We are happy to add an impact statement as we recognize that AVD is inherently dual-use and that it would be beneficial to describe best practice for any further research that uses this work.

---

> > ### Comment · Reviewer_ntEx · 2026-07-09
> > **Response to author rebuttal: planned revisions address my concerns**
> >
> > Thank you for the detailed and constructive response. Your planned changes address my main concerns, and I appreciate how directly you engaged with each point.
> >
> > On the central issue, softening "causally inter-meshed" to "mutually reinforcing" or "co-evolving" resolves my concern, provided the change is carried through consistently in the abstract, the introduction, Section 5, and the Figure 11 caption, so the framing and the evidence match throughout. If you are able to add even a brief temporal signal within the corpus, for example that dataset releases tend to precede shifts in formulation choices, that would let you retain stronger language, but it is optional.
> >
> > On Figure 1, reframing it as "even optimistically biased self-reported numbers fail to show progress" is the right move. I would still encourage replacing the visual "no upward trend" claim with a simple trend test, since it is inexpensive and makes the point rigorous rather than impressionistic.
> >
> > On AIxCC, thank you for confirming the source and that you will verify against primary DARPA materials. Please pay particular attention to the finals count of synthetic vulnerabilities, since public reporting varies on that specific figure and a case study's authority depends on it. Citing the primary source inline for each reported number would fully resolve this.
> > The fixes to the research-question inconsistency, the Section 8 roadmap entry, the copyedit pass, the exclusion rationale for smart contracts and malware, the earlier mention of the key empirical studies, the directional Figure 11, the recommendations checklist, and the dual-use Broader Impact statement all address my remaining points.
> >
> > These are currently described as planned changes. Once a revised version reflecting them is uploaded, I will read it and I expect to be in a position to recommend acceptance. I am happy to look at the revision promptly during the discussion period.

---

### Review · Reviewer_rKSd · 2026-07-15

**Summary Of Contributions:**

This paper surveys 87 influential studies on machine-learning-based automated vulnerability detection (ML4AVD), selected from an initial pool of articles through topic-based, citation-based, and full-text screening. It organizes the literature along six dimensions: problem formulation, input/output granularity, programming-language coverage, datasets, ML approaches, and evaluation metrics.

The paper identifies twelve interconnected “pain points” and argues that choices concerning datasets, task formulations, baselines, and evaluation practices reinforce one another, concentrating the field on binary, function-level vulnerability classification for C/C++ code. It also provides recommendations for addressing these limitations and discusses AIxCC as a recent case study.

**Additional Comments:**

1. Instead of a survey introducing the whole field, this is more like a paper trying to make a point of the current research directions by doing statistics on papers, which I am fine with. But it doesn't read like a survey.

2. Pain point 1: I am not a big fan of the criticism of modeling the problem as binary classification. The author argues that since the vulnerability types help the downstream diagnosis, the detection should be modeled as a multi-classification problem. But I think the detection and explanation/categorization can be orthogonal tasks. Maybe lack of explanation/categorization is a pain point, but technically, I don't think binary classification is one.

3. Pain point 2: similar to PP1, I don't think the same-to-same is the problem. It seems to me that the author is trying to say that detection should be fine-grained under a long-span context.

4. PP3-8 explain that the problems of binary classification, C/C++ only, etc,  are caused by the limited quality of the available datasets.

**Audience:**

Yes

**Audience Explanation:**

The paper should interest researchers working on software/code vulnerability. Several concerns discussed in the paper, including code granularity during detection, code language coverage/data leakage for detection,  and explanation of the detection results, should raise interest in researchers/engineers in related domains.

**Claims And Evidence:**

Yes

**Claims Explanation:**

Generally, yes, but to me, the limited quality/coverage of the benchmark datasets is the core reason for most of the pain points. Other pain points, such as evaluation metrics selection between F-1, AUCROC, FP, etc, are well-known trade-offs in general ML research.

So in general, there is nothing wrong with the claims, but it is also not specific to this domain. See my additional comments later.

**Requested Changes:**

- in the end of intro, the author mentioned two research questions, but I only see one. Clarification or revision should be made for this discrepancy.

- The paper could be substantially improved by reorganizing the identified pain points into orthogonal, conceptually distinct dimensions and presenting them more concisely and informatively. Although the current list contains 12 pain points, many appear to arise from a much smaller set of underlying issues, such as dataset quality and the lack of standardized evaluation protocols. Consolidating overlapping items around these root causes would sharpen the paper’s conceptual structure.

- Correct the $F_2/F_{0.5}$ interpretation on page 25.

The manuscript states that $F_2$ gives higher weight to precision and $F_{0.5}$ gives higher weight to recall. Under the standard definition

$$
F_\beta=(1+\beta^2)\frac{PR}{\beta^2P+R},
$$

the reverse is true: $F_2$ emphasizes recall, while $F_{0.5}$ emphasizes precision. Please also verify references [27] and [83] to determine whether only the explanations were reversed or whether the metrics or references were also interchanged.

---

### Review · Reviewer_RkUW · 2026-07-17

**Summary Of Contributions:**

This paper surveys machine learning methods for automated vulnerability detection using 87 highly cited studies. The authors code the studies along six dimensions: problem formulation, input and detection granularity, programming language, datasets, evaluation metrics, and ML approach. The analysis shows that the field has largely converged on a narrow setup: binary classification of C/C++ code at the function level. The paper then identifies twelve pain points and argues that they continue because datasets, problem formulations, baselines, and metrics reinforce one another. Each pain point is paired with a practical recommendation. The authors also use AIxCC as a case study to examine whether a recent major effort follows these recommendations.

I appreciate that the paper tries to explain why the field appears stuck, rather than simply listing its weaknesses. The distributional analysis is useful, and the two reference tables, including the dataset comparison and the prior survey coverage matrix, are especially helpful.

Two claims gave me some pause. First, the idea that the field is stuck seems to rely on self-reported scores that the paper itself says are difficult to compare. To me, the evidence shows that progress is hard to measure, rather than that progress is absent. Second, I was not fully convinced that the pain points are shown to be “causally inter-meshed.” This claim is mainly supported through reasoning and a diagram, rather than direct causal evidence.

**Audience:**

Yes

**Audience Explanation:**

This is a broad, up-to-date critique of how machine-learning-based vulnerability detection is framed and evaluated, and it pulls a lot of scattered concerns into one place. Researchers working on ML for security, program analysis, or empirical ML methodology would find the distributional picture, the dataset comparison, and the recommendations useful. The AIxCC case study also adds timely relevance.

**Broader Impact Concerns:**

I have no broader impact concerns. As a survey and critique rather than a new method or system, the paper carries limited ethical risk. If anything it leans constructive, as it cautions against using newly-published CVEs as a performance metric and gives attention to test-data leakage and to reward-hacking in agentic settings. I do not think a separate Broader Impact Statement is necessary.

**Claims And Evidence:**

Yes

**Claims Explanation:**

The descriptive core of the paper, including the coding of the 87 studies and the overall picture it provides, is clear and well supported. The recommendations also follow naturally from the findings.

My only concern is that two broader claims seem slightly stronger than the evidence. First, the claim that the field is “not making progress” is based on self-reported scores that the paper itself says are not directly comparable. It may therefore be more accurate to say that progress is difficult to measure, rather than that there has been no progress. Second, the claim that the pain points are “causally” connected is mainly supported by discussion and a diagram, rather than direct causal evidence. The word “causal” therefore seems a little too strong.

**Requested Changes:**

1. (non-critical) It might help to soften the "no progress / field is stuck" framing a little, or to support it more directly. As I read it, that point leans on self-reported scores the paper itself describes as not really comparable, so it may be showing that progress is hard to measure rather than absent. A small rewording could be enough, or a short controlled comparison if the authors want to keep the stronger version.
2. (non-critical) I wonder if the word "causal" in "causally inter-meshed" could be eased a bit, unless it can be shown more directly. The feedback-loop story reads well as reasoning, and I found it convincing, but it seemed to come mainly through argument and the diagram, so something like "mutually reinforcing" might sit a little more comfortably with the evidence. If the authors would like to keep "causal," a bit of direct support for the loops could help.

Minor:
1. If it isn't already shared, it might be nice to release the per-paper coding across the six dimensions and the screening decisions, which would make the survey easier to reproduce and fits the open-science point the paper makes.
2. A short note on how the 87 papers were chosen could be helpful, since the keyword plus citation filter may miss very recent or differently-titled work, which is also where any newer diversification might show up.

---

### Review · Reviewer_ipw2 · 2026-07-17

**Summary Of Contributions:**

This paper surveys 87 influential studies on machine-learning-based vulnerability detection, organizing the literature by task formulation, granularity, programming language, datasets, evaluation metrics, and model type. It identifies twelve recurring problems and argues that common datasets, evaluation practices, and task choices reinforce the field’s narrow focus on function-level binary classification for C/C++. The paper also offers recommendations and uses AIxCC as a recent case study. Overall, the survey covers a useful topic, but the review process needs to be described more clearly, and some of the broader claims should be stated more carefully.

**Additional Comments:**

I found the paper useful, and I believe it could become a strong reference for the area. Its main contribution is bringing together problems that are usually discussed separately. My main concerns are not about the topic or the overall direction, but about whether the review process is sufficiently transparent and whether the strongest conclusions are supported by the selected corpus.

**Audience:**

Yes

**Audience Explanation:**

This paper should interest readers beyond vulnerability detection, especially those working on trustworthy ML, code models, and benchmark design. It brings together familiar issues such as data leakage, class imbalance, reproducibility, and spurious correlations, and shows how they interact across the research pipeline. The AIxCC case study also makes the discussion timely.

**Broader Impact Concerns:**

Automated vulnerability detection is a dual-use area, especially when combined with agents that can also exploit or patch vulnerabilities. The paper does not introduce a new offensive capability, but a brief discussion of responsible disclosure, benchmark release practices, and possible misuse of capable vulnerability-discovery agents would be appropriate.

**Claims And Evidence:**

No

**Claims Explanation:**

The paper raises several well supported concerns, but the review process is not described clearly enough, especially how the 87 papers were screened and coded. Some conclusions also go beyond what this citation filtered corpus can support, and the causal language around Figure 11 is too strong. Figure 1 should be framed as showing that progress is hard to compare, not that the field has made no progress. The paper should also correct the discussion of $F_2$, $F_{0.5}$, CWE severity, and the claims about MCC, Informedness, and Markedness.

**Requested Changes:**

### Critical for acceptance

1. Explain how the 87 papers were screened and coded, including how many reviewers were involved, whether coding was done independently, how disagreements were resolved, and whether the coding data will be released.

2. Soften the causal language. The current evidence shows that the pain points are related and may reinforce one another, but it does not establish causality. Figure 11 should also explain what each edge represents.

3. Reframe Figure 1 as evidence that progress is difficult to compare across papers, rather than evidence that the field has made no progress.

4. Correct the technical statements about $F_2$, $F_{0.5}$, CWE severity, MCC, Informedness, and Markedness.

### Would strengthen the paper

1. Make clear that the main findings describe a citation-filtered set of influential papers, and discuss the possible bias introduced by the search terms and citation threshold.

2. Proofread the paper carefully and consider adding a short table summarizing the evidence and recommendation for each pain point.